# Unbounded Activations for Constrained Monotonic Neural Networks

## Abstract

Monotonic multi-layer perceptrons (MLPs) are crucial in applications requiring interpretable and trustworthy machine learning models. This particularly applies to domains, in which decisions must adhere to specific input-output relationships. Traditional approaches that build monotonic MLPs with universal approximation guarantees often rely on constrained weights and bounded activation functions, which suffer from optimization issues. In this work, we prove that non-negative constrained weights MLPs with activations that saturate on alternating sides are universal approximators for the class of monotonic functions. Given this new result, we show that non-positive constrained weights MLPs with convex monotone activations, contrary to their non-negative constrained counterpart, are universal approximators. Despite such guarantees, we also show that such classes of MLPs are hard to optimize. Therefore, we propose a novel parameterization that eliminates the need for weight constraints. This approach allows the networks to dynamically adjust activations based on weight signs, which enhances optimization stability and performance. Experiments demonstrate that our approach maintains theoretical guarantees and outperforms existing monotonic architectures in approximation accuracy.

## 1 Introduction

Monotonic neural networks represent a pivotal shift in deep learning. They bridge the gap between high-capacity non-linear models and the need for interpretable, consistent outputs in various applications. Monotonic MLPs preserve input-output monotonic relationships, which makes them particularly suitable for domains that require justified and transparent decisions (Gupta et al., 2016; Nguyen & Martínez, 2019). In general, the enforcement of constraints on the model architecture guarantees certain desired properties, such as fairness or robustness.

Furthermore, explicitly designing the model with inductive biases that exploit prior knowledge has been shown to be fundamental for efficient generalization (Dugas et al., 2000; Milani Fard et al., 2016; You et al., 2017). For this reason, the use of monotonic networks can both help for performance improving (Mitchell, 1980) and for data efficiency (Veličković, 2019).

Recent works in this field usually fall into one of the following two categories: 'soft monotonicity' and 'hard monotonicity'. Soft monotonicity employs optimization constraints (Gupta et al., 2019; Sill & Abu-Mostafa, 1996), usually as additional penalty terms in the loss. This class of approaches benefits from its simple implementation and cheap computation. They exploit the power of Multi-Layer Perceptrons (MLPs) to be able to approximate arbitrary functions. However, they often suffer from in-distribution guarantees since penalties are usually applied to dataset samples. For this reason, they struggle to generalize the constraint out-of-distribution. Hard monotonicity instead gives guarantees by construction by imposing constraints in the model architecture (Wehenkel & Louppe, 2019; Nolte et al., 2023). Such guarantees usually come at the cost of effectiveness, leading to vanishing gradient dynamics due to the usage of sigmoid-like activations or dead neuron dynamics due to the usage of ReLU6. The simplest way to do so is to constrain the MLP weights to be non-negative and to use monotonic activations (Daniels & Velikova, 2010). The proposed methods in the literature that exploit this parametrization (Daniels & Velikova, 2010; Wehenkel & Louppe, 2019) require the usage of bounded activations, such as sigmoid and hyperbolic tangent. Even though the usage of bounded activations with MLPs with constrained weights allows proving their universal ap-

proximation abilities for monotonic functions,they are also well-known to be hard to optimize due to vanishing gradients. This shortcoming is even more evident in monotonic NNs with non-negative weights, where bounded activations make the initialization even more crucial for optimization. As discussed in Section 4.1, poor initialization may lead to saturated activations at the beginning of training, thus slowing it down significantly. Indeed, most recent advances in NNs use activations in the family of rectified linear functions, such as the popular `ReLU` activation (Vaswani, 2017; He et al., 2016). However, ReLU activations are problematic for non-negative-weight MLPs as they can only approximate convex monotonic functions (Daniels & Velikova, 2010; Mikulincer & Reichman, 2022). Indeed, any non-negative-weight MLP that uses a convex activation can only approximate convex functions, which severely limits its application. For this reason, many approaches in the literature still rely on including bounded activations like sigmoid, which are known to be universal approximators for the class of monotonic functions (Daniels & Velikova, 2010).

The primary aim of this work is to extend the theoretical basis of monotonic-constrained MLPs, by showing that using activations that saturate on one side, such as ReLU, it is still possible to achieve universal approximation. To showcase that these new findings are not just theoretical tools, we create a new architecture that only uses saturating activations with comparable performances to state-of-the-art architectures. Our contributions can be summarized as follows:

- We show that constrained MLPs that alternate left-saturating and right-saturating monotonic activations are universal approximators for the class of monotonic functions. We also demonstrate that this can be achieved with a constant number of layers, which matches the best-known bound for threshold-activated networks.

- Contrary to the non-negative-constrained formulation, we prove that an MLP with $2n$ layers ($n \geq 2$), non-positive-constrained weights, and ReLU activation is a universal approximator. More generally, this holds true for any saturating monotonic activation.

- We propose a simple parametrization scheme for monotonic MLPs that (i) can be used with saturating activations (ii) does not require constrained parameters, thus making the optimization more stable and less sensitive to initialization (iii) does not require multiple activations, and (iv) does not require any prior choice of alternation of any activation and its point reflection.

Our approach will focus primarily on ReLU activations, which are widely used in the latest advancements in deep learning. However, the results apply to the broader family of monotonic activations that saturate on at least one side. This includes most members of the family of ReLU-like activations such as exponential, ELU (Clevert et al., 2016), SeLU (Klambauer et al., 2017), SReLU (Jin et al., 2016), and many more.

## 2 RELATED WORK

Monotonicity in neural network architectures is an active area of research that has been addressed both theoretically and practically. Prior work can be broadly classified into two categories: architectures designed with built-in constraints (hard monotonicity) and those employing regularization and heuristic techniques to enforce monotonicity (soft monotonicity). Our contribution falls into the former category, but without the complexity that comes from existing methods.

### 2.1 HARD MONOTONICITY

Hard-monotonicity aims at building MLPs with provably monotonicity for any point of the input space. They do so by constructing the MLP so that only monotonic functions can be learned. Initial attempts were Deep Lattice Networks (You et al., 2017) and methods constraining all weights to have the same sign exemplify this approach (Dugas et al., 2009; Runje & Shankaranarayana, 2023; Kim & Lee, 2024). However, constraining the parameters to be non-negative violates the original MLP formulation, thus invalidating the universal approximation theorem. Indeed, the architecture's universal abilities are proven only under the condition that the threshold function is used as activation and that the network is at least 4 layers deep (Runje & Shankaranarayana, 2023). Furthermore, the non-negative parameter constraint also creates issues from an initialization standpoint, as the assumption for popular initializers might be violated for positive semidefinite matrixes.

Only recently, new architectures have been proposed with novel techniques: adapting the Deep Lattice framework to MLPs (Yanagisawa et al., 2022), working with multiple activations (Runje & Shankaranarayana, 2023), constraining the Lipschitz constant (Raghu et al., 2017). However, Runje & Shankaranarayana (2023) requires the usage of multiple activations, and an a priori split of the layer neurons between them, which might be sub-optimal or require additional tuning, and Nolte et al. (2023) relies on very specific activations to control such property, as reported by the authors. In contrast, our work aims to overcome these drawbacks by enhancing flexibility without compromising the monotonicity constraint.

## 2.2 Soft Monotonicity

Soft-monotonicity aims to build monotonic MLPs by working on the training instead of architecture, either using heuristics or regularizations. Techniques such as the point-wise penalty for negative gradients (Gupta et al., 2019; Sill & Abu-Mostafa, 1996) and using Mixed Integer Linear Programming (MILP) for certification (Liu et al., 2020) have been proposed. These methods maintain considerable expressive power but do not guarantee monotonicity. Additionally, the computational expense required for certifications, such as those using MILP or Satisfiability Modulo Theories (SMT) solvers, can be prohibitively high.

## 3 Monotone MLP

A function $f : \mathbb{R}^d \to \mathbb{R}$ is said to be monotone non-decreasing with respect to $x_i$, if given $x_i^0, x_i^1 \in \mathbb{R}, i \in [1, d]$, has the following property:

$$x_i^0 \leq x_i^1 \Rightarrow f(x_1, \ldots, x_i^0, \ldots, x_d) \leq f(x_1, \ldots, x_i^1, \ldots, x_d) \tag{1}$$

And similarly, a function $f : \mathbb{R}^d \to \mathbb{R}$ is said to be monotone non-increasing with respect to $x_i$ if:

$$x_i^0 \leq x_i^1 \Rightarrow f(x_1, \ldots, x_i^0, \ldots, x_d) \geq f(x_1, \ldots, x_i^1, \ldots, x_d). \tag{2}$$

**Observation 1.** *Given $f(x), g(x)$ monotonic non-decreasing, and $h(x), u(x)$ monotonic non-increasing, $f \circ g$ is monotonic non-decreasing, $f \circ h$ is monotonic non-increasing, and $u \circ h$ is monotonic non-decreasing.*

In this work, we will only focus on parametrizing non-decreasing functions, as the monotonicity can be reversed by simply inverting the sign of the inputs[1].

An MLP is defined as a parametrized function $f_\theta$ obtained as composition of alternating affine transformations $l_\theta^{(i)}$ and non-linear activations $\sigma_\theta^{(i)}$:

$$f_\theta(x) = l_\theta^{(1)} \circ \sigma_\theta^{(1)} \ldots \sigma_\theta^{(n-1)} \circ l_\theta^{(n)}. \tag{3}$$

A straightforward approach to building a provable monotonic MLP that respects Equation 1 is to impose constraints on its weights and activations, forcing each term in Equation 3 to be monotonic. Indeed, monotonic activations are, by definition, monotonic. Instead, for affine transformations $l_\theta^{(i)}(x) = W^{(i)}x + b^{(i)}$, we only need to enforce the Jacobian to be non-negative, which is simply the matrix $W^{(i)}$.    FIX  FIX

To optimize the MLP with unconstrained gradient-based approaches, the non-negative weight constraint is obtained using reparametrization, i.e., $l_\theta^{(i)}(x) = g(W^{(i)})x + b^{(i)}$ for some differentiable $g : \mathbb{R} \to \mathbb{R}_+$. Typical reparametrizations use absolute value or squaring.

### 3.1 Known universal approximation conditions

Despite their surprising performances, one critical flaw of existing MLP architectures based on weight constraints is the narrow choice of activation functions. Constrained MLPs have been shown to be universal function approximators for monotonic functions, provided the activation is chosen to be the threshold function, and the number of hidden layers is larger than the dimension of the input variable (Daniels & Velikova, 2010). Just recently, this result has been drastically improved to    FIX

---

[1]In the rest of the paper we will use "monotonic" as shorthand for "monotonic non-decreasing", unless otherwise explicitly specified.

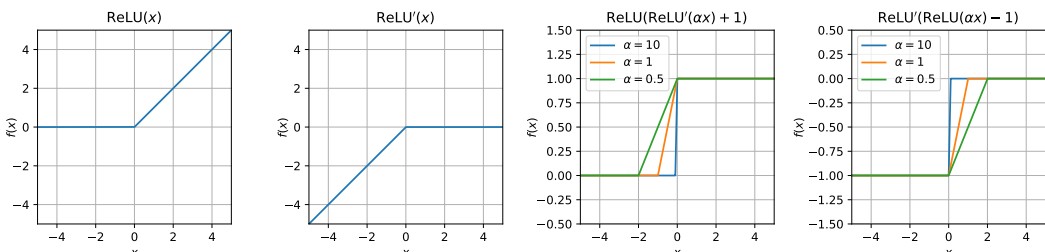

Figure 1: Constructions of Heavyside function using a composition of ReLU and its point reflection ReLU' transformations

a constant bound which proves that four layers are sufficient to have universal approximation properties (Mikulincer & Reichman, 2022). Therefore, practical implementation still resorts to bounded activations such as sigmoid, tanh, or ReLU6. To understand why this is the case, consider:

**Proposition 1.** *The composition of monotonic convex functions is itself monotonic convex.*

Since affine transformations are simultaneously convex and concave, if the activation is chosen to be a monotonic convex function, then the constrained MLP will only be able to approximate monotonic convex functions. Therefore, the use of convex activations like ReLU in a constrained MLP severely limits the expressivity of the network. Despite this clear disadvantage, there is interest in ReLU-activated monotonic MLPs due to their properties and their performances shown in the unconstrained case (Glorot & Bengio, 2010; Hein et al., 2019). Runje & Shankaranarayana (2023) propose a way to introduce RELU activation in the network. The architecture uses multiple activation functions derived from a primitive activation $\sigma(x)$, its point reflection $\sigma'(x) = -\sigma(-x)$, and, in particular, a bounded sigmoid-like activation. While the ReLU activations are ignored in the subsequent theoretical analysis, this last bounded activation function is used to ensure the universal approximation property. Moreover, the bound on the required number of layers is obtained from the result in Daniels & Velikova (2010), which scales linearly with the number of input dimensions. However, we will show that the additional bounded activation is not necessary and that the bound can be improved to the current best known, which is constant with respect to the input dimensions.

### 3.2 UNIVERSAL APPROXIMATION THEOREM FOR NON-THRESHOLD ACTIVATIONS

To have a bound on the minimum number of layers required to guarantee universal approximation with ReLU and its point reflection, we can limit and observe that we can approximate the Heavyside function arbitrarily well with two layers, as shown in Figure 1. With such observation, we can leverage the result of Daniels & Velikova (2010). However, such a bound outlines a linear dependence between input size and the required layers. Further information can be found in Appendix A.1.

Instead, in this section, we will derive a tighter bound, compared to the one proposed by Runje & Shankaranarayana (2023), that instead matches the bound derived in Mikulincer & Reichman (2022) while applying to a broader class of activation functions. This result proves that ReLU-activated constrained MLPs can be as expressive as the logistic variants. More generally, we will show that alternating monotonic activations that saturate on different sides in an MLP are sufficient to ensure universal approximation capabilities with 3 hidden layers.

**Definition 1.** *Given an activation function $\sigma : \mathbb{R} \to \mathbb{R}$, consider:*

$$\sigma(+\infty) \coloneqq \lim_{x \to +\infty} \sigma(x) \qquad\qquad \sigma(-\infty) \coloneqq \lim_{x \to -\infty} \sigma(x). \qquad (4)$$

*We will say that $\sigma$ is right-saturating if $\sigma(+\infty) \in \mathbb{R}$, and it is left-saturating if $\sigma(-\infty) \in \mathbb{R}$, that is, if the corresponding limit exists and is finite. We will denote the set of right-saturating activations as $\mathcal{S}^+$ and the set of left-saturating activations as $\mathcal{S}^-$.*

**Proposition 2.** *For every MLP with non-negative weights and activation $\sigma(x)$, and for any $a \in \mathbb{R}_+, b \in \mathbb{R}$, there exists an equivalent MLP with non-negative weights and activation $a\sigma(x) + b$.*

In the following proofs, thanks to Proposition 2, we will only consider activations that saturate to zero. The main result we will prove is the following:

**Theorem 1.** *An MLP $g_\theta : \mathbb{R}^d \to \mathbb{R}$ with non-negative weights and 3 hidden layers can interpolate any monotonic non-decreasing function $f(x)$ on any set of $n$ points, provided that the activation functions are monotonic non-decreasing and alternate saturation sides. That is, in addition to monotonicity, either of the following holds:*

$$\sigma^{(1)} \in \mathcal{S}^-, \sigma^{(2)} \in \mathcal{S}^+, \sigma^{(3)} \in \mathcal{S}^- \qquad\qquad \sigma^{(1)} \in \mathcal{S}^+, \sigma^{(2)} \in \mathcal{S}^-, \sigma^{(3)} \in \mathcal{S}^+ \qquad (5)$$

The first step is proving that hidden units in the first layer can approximate piecewise constant functions on specific half-spaces.

**Lemma 1.** *Consider an arbitrary hyperplane defined by $\alpha^T (x - \beta) = 0$, $\alpha \in \mathbb{R}_+^d$ and $\beta \in \mathbb{R}^d$, and the open half-spaces $A^+ = \{x : \alpha^T (x - \beta) > 0\}$, $A^- = \{x : \alpha^T (x - \beta) < 0\}$. The $i$-th neuron in the first hidden layer of an MLP with non-negative weights can approximate [2]:*

$$h_i^{(1)}(x) \approx \begin{cases} \sigma^{(1)}(+\infty), & \text{if } x \in A^+ \\ \sigma^{(1)}(-\infty), & \text{if } x \in A^- \\ \sigma^{(1)}(0), & \text{otherwise} \end{cases}$$

*Proof.* Denote by $w$ the weights and by $b$ the bias associated with the hidden unit in consideration. Then, setting the parameters to $w = \lambda \alpha^T$ and $b = \lambda \alpha^T \beta$ and taking the limit we have that:

$$h_i^{(1)}(x) \approx \lim_{\lambda \to +\infty} \sigma^{(1)}(wx + b) = \lim_{\lambda \to +\infty} \sigma^{(1)}\left(\lambda \alpha^T (x - \beta)\right)$$

The limit is either $\sigma^{(1)}(+\infty)$, $\sigma^{(1)}(-\infty)$ or $\sigma^{(1)}(0)$ depending on the sign of $\alpha^T (x - \beta)$, proving that:

$$h_i^{(1)}(x) \approx \begin{cases} \sigma^{(1)}(+\infty), & \text{if } \alpha^T (x - \beta) > 0 \\ \sigma^{(1)}(-\infty), & \text{if } \alpha^T (x - \beta) < 0 \\ \sigma^{(1)}(0), & \text{if } \alpha^T (x - \beta) = 0 \end{cases}$$

$\square$

The second step is proving that one hidden layer can perform intersections of sub-spaces under specific conditions. In our construction, these will be either half-spaces or intersections of specific half-spaces.

**Lemma 2.** *Consider the intersection $A = \bigcap_{i=0}^n A_i$, for $A_1, \ldots, A_n$ subsets of $\mathbb{R}^d$. For any $\gamma$ in the image of $\sigma^{(k)}$, a single unit in the $k$-th hidden layer of an MLP with non-negative weights can approximate:*

$$h_j^{(k)}(x) \approx \gamma \mathbb{1}_A(x)$$

*provided that $h_i^{(k-1)}(x) \approx 0$ for $x \in A_i$, and either:*

- $\sigma^{(k)} \in \mathcal{S}^-$ *and* $h_i^{(k-1)}(x) < 0$ *for* $x \notin A_i$

- $\sigma^{(k)} \in \mathcal{S}^+$ *and* $h_i^{(k-1)}(x) > 0$ *for* $x \notin A_i$

*Proof.* Denote by $w$ the weights and by $b$ the bias associated with the hidden unit in consideration. Then, setting the weights to $w = \lambda \mathbf{1}^T$ and taking the limit we have that:

$$h_j^{(k)}(x) \approx \lim_{\lambda \to +\infty} \sigma^{(k)}(wh^{(k-1)}(x) + b) = \lim_{\lambda \to +\infty} \sigma^{(k)}\left(b + \lambda \sum_{i=0}^n h_i^{(k-1)}(x)\right)$$

Note that in any case, if $x \in \bigcap_{i=1}^n A_i$, then $\lambda \sum_i h_i^{(k-1)}(x) \approx 0$, and the limit simply reduces to $\sigma^{(k)}(b)$. On the other hand, for $x \notin \bigcap_{i=1}^n A_i$, the limit can be either $\sigma^{(k)}(\pm\infty)$ depending on the sign of $h_i^{(k-1)}(x)$. When $\sigma^{(k)} \in \mathcal{S}^-$ and $h_i^{(k-1)}(x) < 0$, the limit is simply $\sigma^{(k)}(-\infty) = 0$. Similarly, when $\sigma^{(k)} \in \mathcal{S}^+$ and $h_i^{(k-1)}(x) > 0$ the limit is $\sigma^{(k)}(+\infty) = 0$.

---

[2] Note that $\sigma^{(1)}(\pm\infty)$ need not be finite.

In both cases, for any $\gamma$ in the image of $\sigma^{(k)}$ we can find a bias value $b$ so that:

$$h_j^{(k)}(x) \approx \gamma \mathbb{1}_A(x) = \begin{cases} \sigma^{(k)}(b) = \gamma, & \text{if } x \in \bigcap_{i=1}^n A_i, \\ \sigma^{(k)}(\pm\infty) = 0, & \text{otherwise} \end{cases}$$

$\square$

Thanks to Lemma 1 and Lemma 2, we can now prove the main result, that is Theorem 1.

*Proof of Theorem 1.* We will only prove the case $\sigma^{(1)} \in \mathcal{S}^-$, $\sigma^{(2)} \in \mathcal{S}^+$, $\sigma^{(3)} \in \mathcal{S}^-$. The proof for the opposite case follows the same structure and is reported in A.2

Assume, without loss of generality, that the points $x_1, \ldots, x_n$ are ordered so that $i_1 < i_2 \implies f(x_{i_1}) \leq f(x_{i_2})$, with ties resolved arbitrarly. We will proceed by construction, layer by layer.

**Layer 1** Since the function to interpolate is monotonic, for any couple of points $i_1 < i_2 : f(x_{i_1}) < f(x_{i_2})$ it is possible to find a hyperplane with non-negative normal, with positive and negative half spaces denoted by $A_{i_2/i_1}^+$ and $A_{i_2/i_1}^-$, such that $x_{i_1} \in A_{i_2/i_1}^-, x_{i_2} \in A_{i_2/i_1}^+$.

Using Lemma 1, we can ensure that it is possible to have:

$$\begin{cases} h_i^{(1)}(x) \approx \sigma^{(1)}(-\infty) = 0, & \text{if } x \in A_{j/i}^- \\ h_i^{(1)}(x) \approx \sigma^{(1)}(+\infty) > 0, & \text{otherwise} \end{cases} \tag{6}$$

**Layer 2** Let us construct the set $A_i^{(2)} = \bigcap_{j:j>i} A_{j/i}^-$. Note that the sets $A_i^{(2)}$ always contain $x_i$ and do not contain any $x_j$ for $j > i$. Using Equation 6, we can apply Lemma 2, which ensures that it is possible to have the following[3]:

$$\begin{cases} h_i^{(2)}(x) \approx 0, & \text{if } x \notin A_i^{(2)} \\ h_i^{(2)}(x) \approx \gamma^{(2)} < 0, & \text{otherwise} \end{cases} \tag{7}$$

**Layer 3** Consider $A_i^{(3)} = \bigcap_{j:j<i} \bar{A}_j^{(2)}$, where $\bar{A}_j^{(2)}$ is the complement of $A_j^{(2)}$. Using Equation 7 we can once again apply Lemma 2, which ensures that it is possible to have the following[4]:

$$h_i^{(3)}(x) \approx \gamma^{(3)} \mathbb{1}_{A_i^{(3)}}(x) \tag{8}$$

Now, we will show that $A_i^{(3)}$ represents a level set, i.e. $x_j \in A_i^{(3)} \iff f(x_j) \geq f(x_i)$. To do so, consider that $\bar{A}_i^{(3)} = \bigcup_{j:j<i} A_j^{(2)}$. Since $x_j \in A_j^{(2)}$, then $x_j \in \bar{A}_i^{(3)}$ for $j < i$. Similarly since $x_j$ is the largest point contained in $A_j^{(2)}$, $\bar{A}_i^{(3)}$ cannot contain $x_i$ or any point larger than $x_i$. This shows that $A_i^{(3)}$ contains exactly the points $\{x_j : f(x_j) \geq f(x_i)\}$.

**Layer 4** To conclude the proof, simply take the weights at the fourth layer to be :

$$w = \left[ \frac{f(x_1) - b}{\gamma^{(3)}}, \frac{f(x_2) - f(x_1)}{\gamma^{(3)}}, \ldots, \frac{f(x_n) - f(x_{n-1})}{\gamma^{(3)}} \right]$$

Since the points are ordered, this ensures that $w$ contains all non-negative terms, when bias term $b$ is taken to be $b \leq f(x_1)$. Defining $f(x_0) = b$, the output of the MLP can be expressed as:

$$g_\theta(x) = w^T h^{(3)}(x) + b = b + \sum_{j=1}^n (f(x_j) - f(x_{j-1})) \mathbb{1}_{A_j^{(3)}}(x) \tag{9}$$

Evaluating Equation 9 at any of the points $x_i$, it reduces to the telescopic sum:

$$g_\theta(x_i) = f(x_1) + \sum_{j=2}^i (f(x_j) - f(x_{j-1})) = f(x_i) \tag{10}$$

Thus proving that the network correctly interpolates the target function. $\square$

---

[3]In this case $\gamma^{(2)} < 0$ since we are considering the case where $\sigma^{(2)}$ saturates right.
[4]In this case $\gamma^{(3)} > 0$ since we are considering the case where $\sigma^{(3)}$ saturates left.

### 3.3 Non-positive constrained monotonic MLP

Consider the simple modification to the standard constrained MLP approach described in Equation 3. However, instead of constraining the weights to be non-negative, they are constrained to be non-positive. This simple modification might seem inconsequential. However, we will show that this structure can create more expressive networks than the original, given an even number of layers or, equivalently, an odd number of hidden layers. Indeed, we will show that a non-positive constrained MLP with 3 hidden layers satisfies the conditions of Theorem 1, as long as the activation function saturates on at least one side. This also includes convex activations like ReLU, which provably do not yield universal approximators in the non-negative constrained weight setting.

FIX

FIX

Note that an MLP defined according to Equation 3 is still monotone for an even number of non-positively constrained layers and monotonic activations, by Observation 1. Therefore, it is still possible to construct provably monotonic networks using non-positive weight constraints.

A first crucial observation is that imposing non-positive constraints in two adjacent layers with an activation function in between is equivalent to imposing non-negative constraints in the two layers and using a point-reflected activation function between them.

**Proposition 3.** *An MLP with $\sigma^{(k)}(x)$ activation at layer $k$, where $W_{ij}^{(k)} \leq 0, W_{ij}^{(k+1)} \leq 0 \,\forall i, j$ is equivalent to an MLP where $W_{ij}^{(k)} \geq 0, W_{ij}^{(k+1)} \geq 0 \,\forall i, j$, and activation at layer $k$ $\sigma'^{(k)}(x) = -\sigma^{(k)}(-x)$.*

From this, it follows that an MLP with an even number of layers, non-positive weights, and activation $\sigma$ at all layers is equivalent to an MLP with non-negative weights that alternate activations between $\sigma'$ and $\sigma$. This equivalence can be achieved using Proposition 3 by "flipping" the weight constraints two layers at a time, which also changes the activations at even-numbered layers from $\sigma$ to $\sigma'$.

The second observation is that both $\sigma$ and $\sigma'$ are monotonic functions but saturate in opposite directions.

**Proposition 4.** *If $\sigma(x)$ is monotonic non-decreasing, then its point reflection $\sigma'(x)$ is also monotonic non-decreasing. If $\sigma(x)$ saturates, then $\sigma'(x)$ also saturates but in the opposite direction.*

Therefore, provided that $\sigma$ is a saturating activation, an MLP with at least 4 layers, non-negative weights, and alternating activation $\sigma$ and $\sigma'$ is a universal monotonic approximator, from Theorem 1. Due to the equivalence in Proposition 3, this also shows that:

FIX

FIX

**Proposition 5.** *If $\sigma \in \mathcal{S}^- \cup \mathcal{S}^+$, an MLP with 4 layers, non-positive weights and activation $\sigma$, is a universal approximator for the class of monotonic functions.*

Similarly, we can apply the observations of this section to show that the structure proposed in Runje & Shankaranarayana (2023) can produce universal monotonic approximators using only point reflections without the need for the third activation class.

While using Theorem 1 allows us to prove that a broad class of constrained MLP architectures are universal monotonic approximators, it does not necessarily translate into MLPs that are easily optimizable. One simple observation that shows how this class of functions is not easily optimizable is considering the computation for an arbitrary input $x$ and a newly initialized MLP with ReLU activation. If $x \geq 0$, then $-|W|x \leq 0$, and thus ReLU$(-|W|x) = 0$. If $x \leq 0$, then $-|W|x \geq 0$, and thus ReLU$(-|W|x) \geq 0$. However, the second layer will saturate for the same reason as before. To allow an efficient and effective optimization, we must carefully tune the bias term to avoid having a 0 gradient. For this reason, we will propose an architecture not influenced by this problem in the following sections.

## 4 Addressing the weight constraint

Historically, the first works that proposed a monotonic neural network formulation relied on the fact that forcing the parameters of the network to be non-negative, specifically the matrixes $W$ in the affine transformations, combined with bounded activations, is a sufficient condition to guarantee that the overall induced function is monotonic (Daniels & Velikova, 2010; Sill & Abu-Mostafa, 1996). Recently, Runje & Shankaranarayana (2023) showed a way to build effective monotonic MLPs with such a technique by exploiting multiple activations. However, even though using con-

strained weights and bounded activation is easy to implement and can be optimized with any unconstrained gradient optimizer, it might lead to vanishing gradient dynamics. Instead, we will show how to address this issue while also tackling the necessity of alternate activations to have universal approximation capabilities, working on the architecture of the MLP.

### 4.1 VANISHING GRADIENT IN CONSTRAINED MLPs WITH BOUNDED ACTIVATIONS

As reported in Section 3, a naive approach to ensure monotonicity is to have monotonic activations and to impose monotonicity to the weights, constraining them to be non-negative. For this reason, such networks' affine transformations are usually parametrized as $l(x) = g(W)x + b$, for some transformation $g : \mathbb{R} \to \mathbb{R}_+$. Note that the bias can be any value, as it is a constant and thus does not affect the gradient.

Such networks employed bounded activations, like sigmoid, tanh, or ReLU6, to have convex-concave activations. This peculiarity makes them very sensitive to initialization and can potentially lead to vanishing gradient dynamics(Glorot & Bengio, 2010). To see why constraining weights to be non-negative exacerbates this condition, consider a monotonic MLP with sigmoidal activations, initialized with random weights according to known, widely used initializers, such as Glorot, where each matrix is sampled from a symmetric distribution around zero with some variance. Instead, biases are initialized to zero, as usually done. Let's assume to use $g(x) = |x|$, but the same reasoning can be applied to any other mapping $g$. At this point, the MLP comprises layers of the following form $\sigma(x) = \sigma(|W|x + b)$. Now, let's consider the second layer of such MLP. Since the first layer has applied the sigmoid activation, then $\sigma^{(1)}(x) \in (0, 1)$. Because of this, $|W^{(2)}|\sigma^{(1)}(x)$ will be a product of all non-negative terms. Therefore, its result can become significantly large. Then, when applying the sigmoid activation of the second layer, it will most likely saturate due to the large positive values returned from the affine transformation. Going on with this reasoning for multiple layers, such behavior will be exacerbated. Appendix A.4 shows one example of such behavior for a very simple function. The same behavior occurs for ReLU6 MLPs, where the gradient might even become exactly 0, and for tanh MLPs if, for example, the dataset is normalized, which is one of the most commonly used data-preprocessing. FIX

One possible solution might beusing BatchNormalization layers (Ioffe, 2015). BatchNorm has already shown its effectiveness in tackling initialization and optimization problems. Indeed, BatchNorm is comprised only of the following transformation: FIX

$$BN(x) = \frac{x - \mathbb{E}[x]}{\sqrt{\mathrm{Var}[x] + \epsilon}} \cdot \gamma + \beta$$

Considering that $\sqrt{\mathrm{Var}[x] + \epsilon} > 0$, forcing $\gamma \geq 0$ by construction, for example, using $\gamma = \mathrm{SoftPlus}(\gamma')$, makes such operation monotonic. Usually, it is initialized as $\beta = 0$ and $\gamma = 1$. For this reason, if used as a pre-activation layer, it might address exploding pre-activation values, standardizing them around zero. However, the investigation of this approach falls out of the scope of this work, and it's left as a future line of research.

### 4.2 RELAXING WEIGHT CONSTRAINTS WITH ACTIVATION SWITCHES

Assuming we used the weight-constrained formulation for the construction proposed in Section 3.3, we would still be left to decide the sequence of activations that should be used for the MLP, which might be unclear or necessitate further hyperparameters-tuning. On the other hand, Observation 1 and Proposition 3 suggest that a monotonically non-increasing operation followed by a second monotonically non-increasing operation ensures that the overall computation in Equation 3 stays monotonic. We will exploit this property to build a monotonic MLP that requires no weight constraint or handpicked activation alternation. FIX

Let's thus consider a single layer of a monotonic ReLU MLP, $f(x) = \sigma(|W|x + b)$, where $\sigma$ can be either $\mathrm{ReLU}(x)$ or $\mathrm{ReLU}'(x)$. Instead of constraining weights, we can separate $W$ in two parts, $W^+ = \max(W, 0)$ and $W^- = \min(W, 0)$. Given these two matrices, we can proceed with two separate affine transformations and use the appropriate activation to even the number of monotonically non-increasing terms in Equation 3. Given such a setting, we can parametrize the whole layer as follows, and we will refer to it as pre-activation formulation:

$$f(x) = \mathrm{ReLU}(W^+ x + b) - \mathrm{ReLU}(W^- x + b) \tag{11}$$

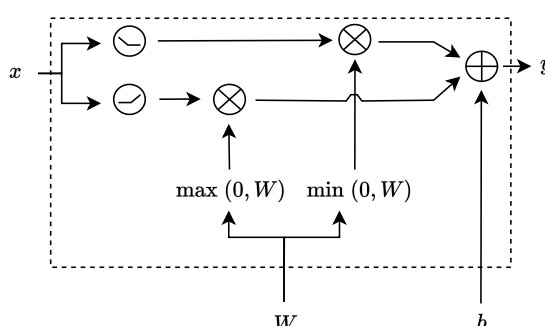

**Algorithm 1** Forward pass of a Monotonic ReLU MLP with post-activation switch

**Input:** data $x \in \mathbb{R}^d$, weight matrix $W \in \mathbb{R}^{d \times d'}$, bias vectors $b \in \mathbb{R}^{d'}$, activation function $\sigma$
**Output:** prediction $\hat{y} \in \mathbb{R}^{d'}$
$W^+ := \max(W, 0)$
$W^- := \min(W, 0)$
$z^+ := W^+ \sigma(x)$
$z^- := W^- \sigma(-x)$
$\hat{y} := z^+ + z^- + b$

Figure 2: On the left, the computation graph of a single layer of a ReLU monotonic NN with the proposed learned activation via weight sign. On the right is the corresponding algorithm pseudocode.

Alternatively, it can be done in the opposite order, from the second layer onwards, and we will call it post-activation formulation:

$$f(x) = W^+ \text{ReLU}(x) + W^- \text{ReLU}(-x) + b \tag{12}$$

In Figure 2, we report only the post-activation switch's pseudocode and computation graph since it will be the formulation that will also be employed for the experimental part of the paper. The pre-activation corresponding algorithm and computational graph can be found in Appendix A.3. Exploiting these formulations, we gain two nice properties. The first one is the relaxation of the weight constraint, no longer needing the $g(W)$ transformation. The second property is that the network can learn the activation it needs by changing the sign of the parameters.Thus, there is no     FIX
need to alternate activations and their point-reflected counterparts manually and, in case, tune such configuration. To conclude, we must show that such a parametrization still allows the MLP to be a universal approximator. To do so, consider a 4-layer MLP with layers formulated as in Equation 12 with $W^{(1)} \geq 0, W^{(2)} \leq 0, W^{(3)} \geq 0, W^{(4)} \leq 0$. Then, just by rearranging the signs from the negative weight matrixes to the activation functions, we end up with a non-negative constrained MLP with alternating activations, thus showing that Theorem 1 holds.

Indeed, the simplicity of the approach can be appreciated: it shares most of the usual steps of the forward pass of an unconstrained MLP and does not require additional special care for initializations. The only additional cost of the proposed method is the double matrix multiplication required by the weight splitting. However, since the two are completely independent, they can be easily performed in parallel.

## 5  EXPERIMENTS

In the previous sections, we studied the proposed method's theoretical properties. In this section,     FIX
we aim to analyze the method's performance compared to other alternatives that give monotonic guarantees. We test our approach starting from the code provided by Runje & Shankaranarayana (2023), guaranteeing a fair comparison of our results and those reported in the original work, which thus will be used as a benchmark for this work. The first dataset used is COMPAS (Fabris et al., 2022). COMPAS is a dataset comprised of 13 features, 4 of which have a monotonic dependency on the classification. A second classification dataset considered is the Heart Disease dataset. It consists of 13 features, 2 of which are monotonic with respect to the output. Lastly, we also test our method on the Loan Defaulter dataset, comprised of 28 features, 5 of which have a monotonic dependency on the prediction. To test on a regression task, we use the AutoMPG dataset comprised of 7 features, 3 of which are monotonically decreasing with respect to the output. A second dataset for regression is the Blog Feedback dataset (Buza, 2013). Contrary to all other datasets, this dataset is composed of a very small portion of monotonic covariates, only accounting for $2.8\%$ of the whole dataset. Indeed, the dataset comprises of 276 features, only 8 of which are monotonic with respect to the output. Furthermore, most of these features are very sparse.

Table 1: Test metrics across different datasets.

| Method | COMPAS (Test Accuracy) | Blog Feedback (Test RMSE) | Loan Defaulter (Test Accuracy) | AutoMPG (Test MSE) | Heart Disease (Test Accuracy) |
|---|---|---|---|---|---|
| Isotonic | 67.6% | 0.203 | 62.1% | - | - |
| XGBoost | $68.5\% \pm 0.1\%$ | $0.176 \pm 0.005$ | $63.7\% \pm 0.1\%$ | - | - |
| Certified | $68.8\% \pm 0.2\%$ | $0.159 \pm 0.001$ | $65.2\% \pm 0.1\%$ | - | - |
| COMET | - | - | - | $8.81 \pm 1.81$ | $86\% \pm 3\%$ |
| DLN | $67.9\% \pm 0.3\%$ | $0.161 \pm 0.001$ | $65.1\% \pm 0.2\%$ | $13.34 \pm 2.42$ | $86\% \pm 2\%$ |
| Min-Max Net | $67.8\% \pm 0.1\%$ | $0.163 \pm 0.001$ | $64.9\% \pm 0.1\%$ | $10.14 \pm 1.54$ | $75\% \pm 4\%$ |
| Constrained MNN | $69.2\% \pm 0.2\%$ | $0.154 \pm 0.001$ | $\mathbf{65.3\% \pm 0.1\%}$ | $8.37 \pm 0.08$ | $89\% \pm 0\%$ |
| Scalable MNN | $\mathbf{69.3\% \pm 0.9\%}$ | $\mathbf{0.150 \pm 0.001}$ | $65.0\% \pm 0.1\%$ | $\mathbf{7.44 \pm 1.20}$ | $88\% \pm 4\%$ |
| **Ours** | $\mathbf{69.3}\% \pm \mathbf{0.3}\%$ | $0.158 \pm 0.001$ | $\mathbf{65.3}\% \pm \mathbf{0.1}\%$ | $7.67 \pm 1.73$ | $\mathbf{94}\% \pm \mathbf{3}\%$ |

We compare our method with several other approaches that give monotonic guarantees by construction. In particular, we compare it to XGBoost(Chen & Guestrin, 2016) as a baseline, Deep Lattice Network (You et al., 2017), Min-Max Networks (Daniels & Velikova, 2010), Certified Networks (Liu et al., 2020), COMET (Sivaraman et al., 2020), Constrained Monotonic Neural Networks (Runje & Shankaranarayana, 2023), and Scalable Monotonic Neural Networks (Kim & Lee, 2024). In Table 1, we report the final test set metrics, comparing the proposed methods with the results obtained from Runje & Shankaranarayana (2023), with missing entries for metrics not reported by the authors. We employed an MLP as shown by Runje & Shankaranarayana (2023), composed of 3 layers for non-monotonic features and 4 subsequent monotonic layers, except for the Blog Feedback dataset, for which smaller layers have been used, to avoid overfitting. Specifically, the post-activation formulation reported in Algorithm 1 has been used for all results. The proposed method matches or surpasses the performances of other recently proposed approaches, except for the case of the Blog Feedback dataset. Such results are obtained with minimal modifications to the architecture used by Runje & Shankaranarayana (2023). However, for the Blog Feedback dataset, given the small number of monotonic features compared to the overall dataset, the performances on this dataset might be influenced more by the architecture or regularization than the inductive bias induced by the monotonic layers. Indeed, running the benchmark only considering the monotonic feature still leads to an average $0.160 \pm 0.001$ RMSE.

# 6 CONCLUSIONS AND FUTURE WORKS

In this work, we proved that MLPs with non-negative constrained weights and alternating activations that saturate at least on one side are universal approximators for the class of monotonic functions. In addition, we show that a specific case of such a setting is defined by a network with monotonic convex activation and constrained non-positive weights. We then use this result to present a new parametrization that relaxes the need for activation alternation and weight constraint while still allowing for monotonic convex activation, which was impossible earlier. With this parametrization, the layer can choose which activation to use based on the parameters' signs. We then use our monotone fully connected layer to build MLPs, we show that we can achieve state-of-the-art performances. Even though this work proves that any monotonic saturating activation can be used to build monotonic MLPs, it's still an open question whether non-saturating activations, such as Leaky-ReLU, can be used to build monotonic MLPs. After that, activation must only be monotonic to be used in monotonic MLPs. Furthermore, batch normalization has proven highly effective in the unconstrained case. Still, it has never been used as a possible solution to the initialization problem for the monotonic case.

FIX

FIX

FIX

# 7 ETHICAL CONSIDERATIONS

NEW

The use of the COMPAS dataset in this research acknowledges its status as a common benchmark within the field of machine learning fairness studies (Angwin et al., 2022; Dressel & Farid, 2018). Recognizing the complexities and potential ethical challenges associated with such datasets, we emphasize a commitment to responsible research practices. We prioritize transparency and ethical rigor throughout our study to ensure that the methodologies employed and the conclusions drawn contribute constructively to the ongoing discourse in AI ethics and fairness. This approach underlines our dedication to advancing machine learning applications in a manner that is conscious of their broader societal impacts.

# 8 REPRODUCIBILITY

In the Appendix A.6 and A.5, we report all necessary information for reproducibility of the results in Table 1 and further information on the dataset employed in this work. Furthermore, the code used to obtain the results in Table 1 is provided.

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

# A APPENDIX

The appendix is structured in the following way:

- **Section A.1**: in this section, we show the arguably simplest though loosest bound to prove that non-negative constrained MLPs with ReLU and ReLU' activations are universal approximators.

- **Section A.2**: in this section we prove the results of Theorem 1 for the opposite alternation case.

- **Section A.3**: as reported in Section 4, we propose two possible parametrizations, a pre-activation switch, and a post-activation switch. In Appendix A.3, the pseudocode and the computational graph of the two can be found.

- **Section A.4**: in this section we will compare the proposed method to the bounded-activation counterpart, showing how the formulation with sigmoidal activation suffers from vanishing gradients.

- **Sections A.5 and A.6**: in these sections, we report further information regarding how the results have been obtained and about the datasets employed for this work.

- **Section A.7**: since Theorem 1 only requires the non-linearity to be saturating, in this section we report a brief overview of other activations that can be applied with the proposed method, in order underline how it is more general than just using ReLU activations.

- **Section A.8**: the proof provided for Theorem 1 is different to the ones previously proposed in literature. However, it still ends with the result of requiring 4 layers to be a universal approximator, as previously shown in Mikulincer & Reichman (2022) for the heavy-side function. For readers that are already familiar with such proof, we also report in Appendix A.8 a proof very similar to the one in Mikulincer & Reichman (2022), trying to reuse as much as possible the original structure.

## A.1 NAIVE BOUND FOR UNIVERSAL APPROXIMATION OF ALTERNATING MLPs

A simpler, though looser, bound to prove that MLPs with alternating ReLU and its point reflection ReLU' activations are a universal monotonic function approximator can be achieved building on the proof of Mikulincer & Reichman (2022). Two simple observations are sufficient.

**Remark 1.** *the composition of ReLU and its point reflections $ReLU'(x) = -ReLU(-x)$ can approximate the threshold function $\mathbb{1}_{x \geq 0}$ arbitrarily well:*

$$\lim_{\alpha \to +\infty} ReLU(ReLU'(\alpha x) + 1) = \mathbb{1}_{x \geq 0}(x) \tag{13}$$

$$\lim_{\alpha \to +\infty} ReLU'(ReLU(\alpha x) - 1) = \mathbb{1}_{x \geq 0}(x) - 1 \tag{14}$$

A representation of Equation 13 is provided in Figure 1.

The reason why we can approximate non-convex functions using only ReLU-like activations is reported in Proposition 1. However, considering Observation 4, we can see how this limitation can be addressed.

**Remark 2.** *The formulas in Equation 13 can be implemented with a 2-layer constrained MLP, alternating ReLU and ReLU' activations.*

This is enough to leverage the existing results for threshold threshold-activated MLP (Mikulincer & Reichman, 2022). This includes the best-known bound on the number of the required hidden layers, which, however, doubles from 3 to 6 due to the need for two ReLU layers for the Heavyside approximation. However, this naive bound is unnecessarily loose, as shown in Theorem 1.

## A.2 Proof for opposite alternation of activation for Theorem 1

In this section, we will conclude the proof of Theorem 1, considering the case with activations that alternate in the opposite direction than the one reported in the main text. Indeed, in Section 3.2 we proved the result for the case with $\sigma^{(1)} \in \mathcal{S}^-, \sigma^{(2)} \in \mathcal{S}^+, \sigma^{(3)} \in \mathcal{S}^-$, while in this section we will prove the case with $\sigma^{(1)} \in \mathcal{S}^+, \sigma^{(2)} \in \mathcal{S}^-, \sigma^{(3)} \in \mathcal{S}^+$. The proof is extremely similar, with just a few opposite signs due to the opposite alternation. Thus, most constructions will be shared.

*Proof of Theorem 1 with opposite alternation.* Assume, without loss of generality, that the points $x_1, \ldots, x_n$ are ordered so that $i_1 < i_2 \implies f(x_{i_1}) \le f(x_{i_2})$, with ties resolved arbitrarly. We will proceed by construction, layer by layer.

**Layer 1** Since the function to interpolate is monotonic, for any couple of points $i_1 < i_2 : f(x_{i_1}) < f(x_{i_2})$ it is possible to find a hyperplane with non-negative normal, with positive and negative half spaces denoted by $A^+_{i_2/i_1}$ and $A^-_{i_2/i_1}$, such that $x_{i_1} \in A^+_{i_2/i_1}, x_{i_2} \in A^+_{i_2/i_1}$.

Using Lemma 1, we can ensure that it is possible to have:

$$\begin{cases} h_i^{(1)}(x) \approx \sigma^{(1)}(+\infty) = 0, & \text{if } x \in A^+_{j/i} \\ h_i^{(1)}(x) \approx \sigma^{(1)}(-\infty) < 0, & \text{otherwise} \end{cases} \tag{15}$$

**Layer 2** Let us construct the set $A_i^{(2)} = \bigcap_{j:j<i} A^+_{i/j}$. Note that the sets $A_i^{(2)}$ always contain $x_i$ and do not contain any $x_j$ for $j < i$. Using Equation 15, we can apply Lemma 2, which ensures that it is possible to have the following[5]:

$$\begin{cases} h_i^{(2)}(x) \approx 0, & \text{if } x \in A_i^{(2)} \\ h_i^{(2)}(x) \approx \gamma^{(2)} > 0, & \text{otherwise} \end{cases} \tag{16}$$

**Layer 3** Consider $A_i^{(3)} = \bigcap_{j:j>i} \bar{A}_j^{(2)}$, where $\bar{A}_j^{(2)}$ is the complement of $A_j^{(2)}$. Using Equation 16 we can once again apply Lemma 2, which ensures that it is possible to have the following[6]:

$$h_i^{(3)}(x) \approx \gamma^{(3)} \mathbb{1}_{A_i^{(3)}}(x) \tag{17}$$

Now, we will show that $A_i^{(3)}$ represents a level set, i.e. $x_j \in A_i^{(3)} \iff f(x_j) \le f(x_i)$. To do so, consider that $\bar{A}_i^{(3)} = \bigcup_{j:j>i} A_j^{(2)}$. Since $x_j \in A_j^{(2)}$, then $x_j \in \bar{A}_i^{(3)}$ for $j > i$. Similarly since $x_j$ is the smallest point contained in $A_j^{(2)}$, $\bar{A}_i^{(3)}$ cannot contain $x_i$ or any point smaller than $x_i$. This shows that $A_i^{(3)}$ contains exactly the points $\{x_j : f(x_j) \le f(x_i)\}$.

**Layer 4** To conclude the proof, simply take the weights at the fourth layer to be :

$$w = \left[ \frac{f(x_1) - f(x_2)}{\gamma^{(3)}}, \ldots, \frac{f(x_{n-1}) - f(x_n)}{\gamma^{(3)}}, \frac{f(x_n) - b}{\gamma^{(3)}} \right]$$

Note that compared to Equation 8, here $\gamma^{(3)}$ is now negative, and the terms in the numerators' difference are reversed. Since the points are ordered, this ensures that $w$ contains all non-negative terms, when bias term $b$ is taken to be $b \ge f(x_n)$. Defining $f(x_{n+1}) = b$, the output of the MLP can be expressed as:

$$g_\theta(x) = w^T h^{(3)}(x) + b = b + \sum_{j=1}^{n} (f(x_j) - f(x_{j+1})) \mathbb{1}_{A_j^{(3)}}(x) \tag{18}$$

Evaluating Equation 18 at any of the points $x_i$, it reduces to the telescopic sum:

$$g_\theta(x_i) = f(x_n) + \sum_{j=i}^{n-1} (f(x_j) - f(x_{j+1})) = f(x_i) \tag{19}$$

Thus proving that the network correctly interpolates the target function. □

---

[5]In this case $\gamma^{(2)} > 0$ since we are considering the case where $\sigma^{(2)}$ saturates left.
[6]In this case $\gamma^{(3)} < 0$ since we are considering the case where $\sigma^{(3)}$ saturates right.

### A.3 ALGORITHMS

In Section 4, we show how we can parametrize the activation switch using the sign of the weights. For such a mechanism, we propose two different parametrizations, one where the switch is applied post-activation and another one pre-activation. In Figure 2, we report both the pseudo-code and the computational graph for the post-activation formulation. For completeness, in this section, we also report the pre-activation pseudo-code and computational graph, and for readability and to ease the comparison, we report them side by side, reporting again the post-activation formulation also reported in the main text. In particular, in Figure 3, we report the two pseudocode side-by-side, and in Algorithm A.3, the relative pseudocodes.

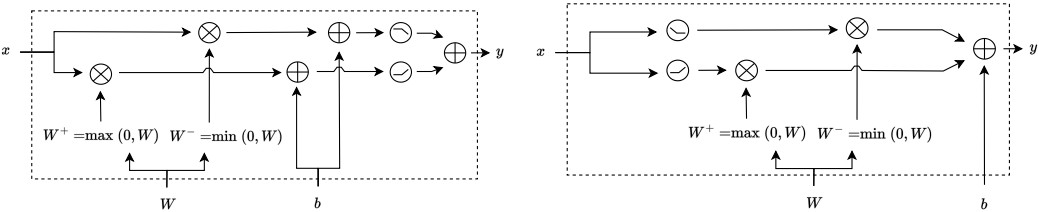

Figure 3: Computation graph of a single layer of a ReLU monotonic NN with the proposed learned activation via weight sign. The left plot reports the computational graph of the pre-activation, and the right plot shows the post-activation switch.

| **Algorithm 2** Forward pass of a Monotonic ReLU MLP with pre-activation switch | **Algorithm 3** Forward pass of a Monotonic ReLU MLP with post-activation switch |
|---|---|
| **Input:** data $x \in \mathbb{R}^n$, weight matrix $W \in \mathbb{R}^{h_l \times h_{l-1}}$, bias vectors $b \in \mathbb{R}^{h_l}$, activation function $\sigma$ | **Input:** data $x \in \mathbb{R}^n$, weight matrix $W \in \mathbb{R}^{h_l \times h_{l-1}}$, bias vectors $b \in \mathbb{R}^{h_l}$, activation function $\sigma$ |
| **Output:** prediction $\hat{y} \in \mathbb{R}^{h_L}$ | **Output:** prediction $\hat{y} \in \mathbb{R}^{h_L}$ |
| $W^+ := \max(W, 0)$ | $W^+ := \max(W, 0)$ |
| $W^- := \min(W, 0)$ | $W^- := \min(W, 0)$ |
| $z^+ := W^+ x + b$ | $z^+ := W^+ \sigma(x)$ |
| $z^- := W^- x + b$ | $z^- := W^- \sigma(-x)$ |
| $\hat{y} := \sigma(z^+) - \sigma(z^-)$ | $\hat{y} := z^+ + z^- + b$ |

### A.4 TOY EXAMPLE

To showcase the effectiveness of the proposed method to the bounded-activation counterpart, in Figure 4 we compare them on a simple synthetic example. In particular, the models are asked to approximate $f(x) = \cos(x) + x$, a simple 1D monotonic function with multiple saddle points. For this reason, it is fundamental for the approximation model to be very flexible. To showcase the different performances, we will test 4 models. The first model to test is an unconstrained NN, which shows that an unconstrained model can learn such a function. The second model is a monotonic NN with non-negative and ReLU activations, which shows that, as shown in theory, it cannot approximate nonconvex function. The third model is a monotonic NN with non-negative and sigmoid activations. This model, instead, is shown to be a universal approximator for monotonic functions but suffers from vanishing gradients. Lastly, the fourth model is the proposed parametrization, specifically the post-activation setting, as described in Section 4.2.

In Figure 4 can be seen how the model with non-negative and ReLU activations cannot learn the function as predicted by theory since the function that is asked to learn is non-convex. Instead, both the sigmoid model and our proposed approach successfully approximate it. Still, the sigmoid function struggles to be optimized due to the complications of using sigmoid activations. Instead, the proposed method exploits rectified linear activations, which, under a regime where the number of dead neurons is not too high, is much easier to optimize, as explained in the original work that introduced such activation Glorot & Bengio (2010) and Raghu et al. (2017).

Such difference is also evident in analyzing the Negative Log Likelihood (NLL) loss of the training. We report in Figure 4 the various training losses obtained with two different sizes of layers. The naive monotonic ReLU, which cannot approximate such a function, is indeed the worst. However, even though the sigmoid monotonic NN is a universal approximator, it is the slowest to learn, probably due to the vanishing gradient problem. Instead, the proposed method that uses ReLU activations is the fastest to converge, almost catching the unconstrained model in the setting with more neurons. Generally speaking, as also reported at the end of Section 3.3, MLPs with constrained weights,

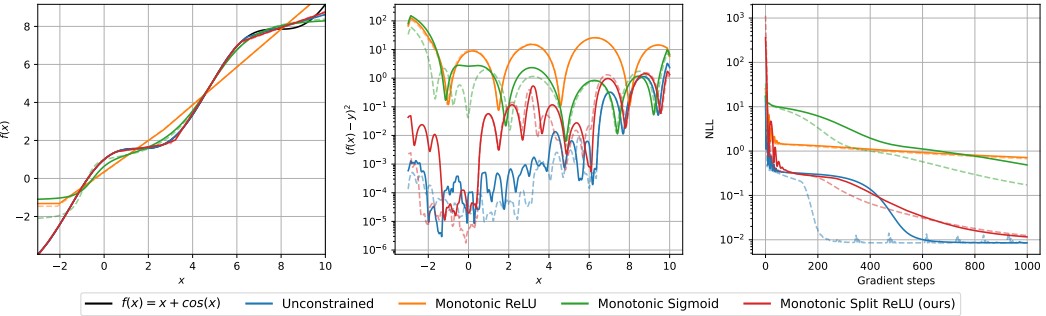

Figure 4: First plot, approximation of $f(x)$ using MLPs with layers of 128 neurons. Second plot, approximation of $f(x)$ using MLPs with layers of 256 neurons. Last plot, training losses of the different methods (full lines represent versions with 128 neurons, dashed lines represent versions with 256 neurons).

require a careful initialization to avoid non-optimizable configurations. The proposed method in Section A.3 alleviates this behavior but is not indifferent to it.

In order to showcase the vanishing gradient problem exacerbated by the non-negatively constraining, in Figure 5 we create a 128-neuron wide MLP with varying numbers of hidden layers, and we compare the average gradient of the output with respect to the parameters on the same function approximation problem presented earlier in Figure 4. It can be observed how the sigmoid monotonic MLP, even with a small number of layers, has one order of magnitude less gradient magnitude; in particular, it has an average gradient of $0.0019$ for 4 layers and $0.00099$ for 10 layers. Instead, the ReLU monotonic MLP has an exploding gradient due to the accumulation of activations induced by the pairing of ReLU-activation and positive weight; in particular, it starts from a gradient magnitude of $3.54$ for 4 layers and goes to $3311.00$ for 10 layers. Finally, the proposed approach keeps the

gradient magnitude in a reasonable magnitude range, starting from a gradient of $0.010$ for 4 layers and going to $1.259$ for 10 layers. Results are averaged over 20 different random initializations, and plot shows $\pm 1\sigma$. In order to better analyze the optimization problems of these architectures, we also report in Figure 6 the distributions of the gradients of a 6-layers MLP with the various architectures. It can be seen that the sigmoid MLP has extremely low gradients for the initial layers, leading to slow learning. On the other hand, the ReLU MLP has exploding gradients for the final layers.      NEW

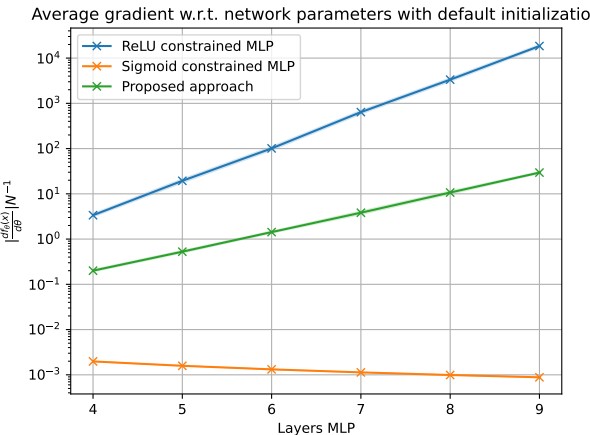

Figure 5: Average gradient from monotonic MLPs varying the number of layers. Data is shown in the log scale for the $y$-axis.

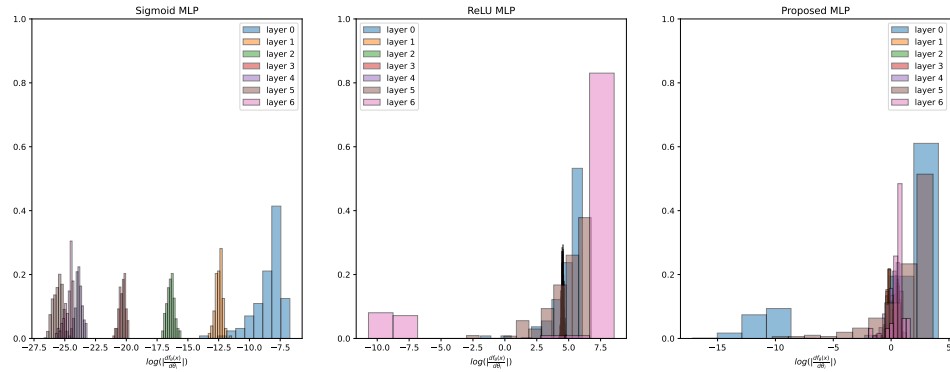

Figure 6: Distribution of gradients from monotonic MLPs for each layer (layer 0 is the final one, layer 6 is the first after the input).

## A.5 Dataset description

For this work, the code was heavily based on the code provided by Runje & Shankaranarayana (2023) in order to ensure that the used dataset matched exactly. For this reason, we will report a short description of the employed dataset, but for a further and more detailed description, refer to the original work (Runje & Shankaranarayana, 2023).

- **COMPAS**: This dataset is a binary classification dataset composed of criminal records, comprised of 13 features, 4 of which are monotonic.
- **Blog Feedback**: This dataset is a regression dataset comprised of 276 features, 8 of which are monotonic, aimed at predicting the number of comments within 24h.
- **Auto MPG**: This dataset is a regression dataset aimed at predicting the miles-per-gallon consumption and is comprised of 7 features, 3 of which are monotonic.
- **Heart Disease**: This dataset is a classification dataset composed of 13 features, 2 of which are monotonic, aimed at predicting a possible heart disease.
- **Loan Defaulter**: This dataset is a classification dataset composed of 28 features, 5 of which are monotonic, and is aimed at predicting load defaults.

## A.6 Experiments description

Following are the specifications used to obtain the results reported in Table 2. The experiments were developed in PyTorch (version 2.4.0). The training was performed using the Adam optimizer implementation from the PyTorch Library. The MLPs comprised 4 layers of PyTorch `Linear`, followed by 4 monotonic layers built with the post-activation proposed method, as reported in Algorithm 1. All non-linear activations used were ReLU, except for the classification datasets where sigmoid was used for the last layer. No hyperparameter tuning was performed except for the Blog FeedBack dataset, for which we used a similar architecture employed in Runje & Shankaranarayana (2023) due to severe overfitting in the non-monotonic section of the MLP.

Table 2: Hyper-parameters used for results reported in Table 1

| Hyper-parameter | COMPAS | Blog Feedback | Loan Defaulter | AutoMPG | Heart Disease |
|---|---|---|---|---|---|
| Learning-rate | $10^{-4}$ | $10^{-2}$ | $10^{-4}$ | $10^{-3}$ | $10^{-4}$ |
| Epochs | 50 | 50 | 50 | 50 | 50 |
| Batch-size | 256 | 256 | 256 | 8 | 8 |
| Free layers size | 32 | 4 | 32 | 16 | 32 |
| Monotonic layers size | 32 | 12 | 32 | 16 | 32 |

## A.7 Extension to other activations

In the rest of the paper, for all the practical examples, we assumed that ReLU was the activation chosen for the MLP. However, the results in Sections 3.3 and 4.2 only require that the activation function saturates in at least one of the two sides, other than being monotonic. If ReLU falls in such a category, it is not the only one, and many other widely used ReLU-like activations satisfy the minimal assumptions of Theorem 1. For this reason, we will now analyze many other activations and report whether they comply with our construction. In particular, we report in Table 3 multiple widely used activations. With them, we also report the respective gradients, whether they are non-decreasing and saturating, and whether they can be used for the proposed approach.

It can be seen that the proposed method allows the usage of most of today's widely used activations. However, it is crucial to notice that even though the proposed method allows for saturating activations, it also can be used with bounded activations, such as sigmoid and tanh, but that might bring almost no additional advantage over the weight-constrained counterpart. Any activation that saturates at least one side can be used, given that it is monotonic. Still, the real advantage comes from activations that saturate only one side.

Table 3: Widely used activations with corresponding their properties, and whether they can be used or not.

| Name | Function | Gradient | Monotone | Saturates | Usable |
|---|---|---|---|---|---|
| ReLU | $\begin{cases} x & \text{if } x \geq 0 \\ 0 & \text{otherwise} \end{cases}$ | $\begin{cases} 1 & \text{if } x \geq 0 \\ 0 & \text{otherwise} \end{cases}$ | ✓ | ✓ | ✓ |
| LeakyReLU | $\begin{cases} x & \text{if } x \geq 0 \\ \alpha x & \text{otherwise} \end{cases}$ | $\begin{cases} 1 & \text{if } x \geq 0 \\ \alpha & \text{otherwise} \end{cases}$ | ✓[1] | ✗ | ✗ |
| PReLU | $\begin{cases} x & \text{if } x \geq 0 \\ \alpha x & \text{otherwise} \end{cases}$ ($\alpha$ learnable) | $\begin{cases} 1 & \text{if } x \geq 0 \\ \alpha & \text{otherwise} \end{cases}$ | ✓[1] | ✓ | ✓[1] |
| ReLU6 | $\begin{cases} 6 & \text{if } x \geq 6 \\ x & \text{if } 0 \leq x \leq 6 \\ 0 & \text{otherwise} \end{cases}$ | $\begin{cases} 0 & \text{if } x \geq 6 \\ 1 & \text{if } 0 \leq x \leq 6 \\ 0 & \text{otherwise} \end{cases}$ | ✓ | ✓ | ✓ |
| ELU | $\begin{cases} x & \text{if } x \geq 0 \\ \alpha(e^x - 1) & \text{otherwise} \end{cases}$ | $\begin{cases} 1 & \text{if } x \geq 0 \\ \alpha e^x & \text{otherwise} \end{cases}$ | ✓[1] | ✓ | ✓[1] |
| SELU | $\lambda \begin{cases} x & \text{if } x \geq 0 \\ \alpha(e^x - 1) & \text{otherwise} \end{cases}$ | $\lambda \begin{cases} 1 & \text{if } x \geq 0 \\ \alpha e^x & \text{otherwise} \end{cases}$ | ✓[1] | ✓ | ✓[1] |
| GeLU | $x\Phi(x)$ | $\Phi(x)\frac{1}{\sqrt{2\pi}}e^{\frac{-x^2}{2}}$ | ✗ | ✓ | ✗ |
| SiLU/Swish | $x\sigma(x)$ | $\frac{e^x(x+e^x+1)}{(e^x+1)^2}$ | ✗ | ✓ | ✗ |
| Sigmoid | $\frac{1}{1+e^{-x}}$ | $\frac{e^{-x}}{(1+e^{-x})^2}$ | ✓ | ✓ | ✓ |
| Tanh | $\frac{e^x-e^{-x}}{e^x+e^{-x}}$ | $1 - \left(\frac{e^x-e^{-x}}{e^x+e^{-x}}\right)^2$ | ✓ | ✓ | ✓ |
| Exp | $e^x$ | $e^x$ | ✓ | ✓ | ✓ |
| SoftSign | $\frac{x}{|x|+1}$ | $\frac{1}{(|x|+1)^2}$ | ✓ | ✓ | ✓ |
| Softplus | $\log(1+e^x)$ | $\frac{e^x}{e^x+1}$ | ✓ | ✓ | ✓ |
| LogSigmoid | $-\log(1+e^{-x})$ | $\frac{1}{1+e^x}$ | ✓ | ✓ | ✓ |

[1]: true only if parametrized in such a way to guarantee $\alpha \geq 0$

## A.8 ALTERNATIVE PROOF OF THEOREM 1

In this section, we will construct a proof similar to the one proposed by Mikulincer & Reichman (2022) to prove the constant bound of required layers for a constrained MLP with Heavyside activations.

**First layer construction**   First, let us show that the network can represent piece-wise functions at the first hidden layer.

**Lemma 3.** *Consider an hyperplane defined by $\alpha^T (x - \beta) = 0$, $\alpha \in \mathbb{R}^k_+$ and $\beta \in \mathbb{R}^k$, and the open half-spaces:*

$$A^+ = \{x : \alpha^T (x - \beta) > 0\}, \tag{20}$$
$$A^- = \{x : \alpha^T (x - \beta) < 0\}. \tag{21}$$

*A single neuron in the first hidden layer of an MLP with non-negative weights can approximate [7]:*

$$h^{(1)}(x) \approx \begin{cases} \sigma^{(1)}(+\infty), & \text{if } x \in A^+ \\ \sigma^{(1)}(-\infty), & \text{if } x \in A^- \\ \sigma^{(1)}(0), & \text{otherwise} \end{cases}$$

*Proof.* Denote by $w$ the weights and by $b$ the bias associated with the hidden unit in consideration. Then, for any $\lambda \in \mathbb{R}_+$, setting the parameters to $w = \lambda\alpha^T$ and $b = \lambda\alpha^T\beta$ we have that:

$$h = \sigma^{(1)}(wx + b) = \sigma^{(1)}\left(\lambda\alpha^T (x - \beta)\right)$$

in the limit, we get:

$$h^{(1)}(x) \approx \lim_{\lambda \to +\infty} \sigma^{(1)}\left(\lambda\alpha^T (x - \beta)\right)$$

The limit is either $\sigma^{(1)}(+\infty)$, $\sigma^{(1)}(-\infty)$ or $\sigma^{(1)}(0)$ depending on the sign of $\alpha^T (x - \beta)$, proving that

$$h^{(1)}(x) \approx \begin{cases} \sigma^{(1)}(+\infty), & \text{if } \alpha^T (x - \beta) > 0 \\ \sigma^{(1)}(-\infty), & \text{if } \alpha^T (x - \beta) < 0 \\ \sigma^{(1)}(0), & \text{if } \alpha^T (x - \beta) = 0 \end{cases}$$

$\square$

For an easier interpretation of the just state construction, we show in Figure 7 some samples from the family of functions that can be learned with this first hidden layer.

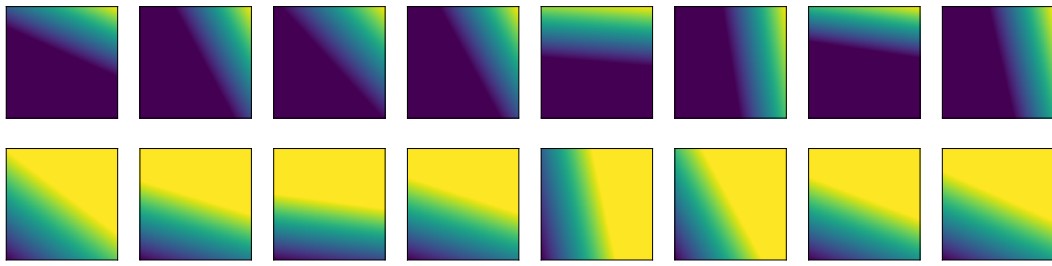

Figure 7: Examples of learnable functions at the first hidden layer.

---

[7]Note that $\sigma^{(1)}(\pm\infty)$ needs not be finite.

**Second layer construction** Using Lemma 3, we can show that alternating saturation directions in the activations is sufficient to represent indicator functions of intersections and unions of positive half-spaces.

**Lemma 4.** *If $\sigma^{(1)} \in \mathcal{S}^+, \sigma^{(2)} \in \mathcal{S}^-$, there exists a rescaling factor $\gamma \in \mathbb{R}_+$ such that a single unit in the second hidden layer of an MLP with non-negative weights, can approximate:*

$$h^{(2)}(x) \approx +\gamma \mathbb{1}_{A^\cap}(x)$$

*for any $A^\cap = \bigcap_{i=1}^n A_i^+$.*

*Similarly, if $\sigma^{(1)} \in \mathcal{S}^-, \sigma^{(2)} \in \mathcal{S}^+$, it can approximate*

$$h^{(2)}(x) \approx +\gamma \mathbb{1}_{A^\cup}(x) - \gamma$$

*for any $A^\cup = \bigcup_{i=1}^n A_i^+$.*

*Proof.* Denote by $w$ the weights and by $b$ the bias associated to the hidden unit in consideration at the second layer. For any $\lambda \in \mathbb{R}_+$ , setting the weights to $w = \lambda \mathbf{1}^T$ we have that

$$h^{(2)}(x) = \sigma^{(2)}\left(wh^{(1)} + b\right) = \sigma^{(2)}\left(b + \lambda \sum_i h_i^{(1)}\right)$$

Taking the limit, the result only depends on the sign of $\sum_i h_i^{(1)}$. Using Lemma 3, we can ensure that it is possible to have

$$h_i^{(1)}(x) \approx \begin{cases} \sigma^{(1)}(+\infty), & \text{if } x \in A_i^+ \\ \sigma^{(1)}(-\infty), & \text{if } x \in A_i^- \end{cases}$$

From here, there are two cases, depending on the saturation of the activations. We will only prove the case when the activations saturate to zero to avoid needlessly complicated formulas. However, the result holds even in the general case.

If we assume $\sigma^{(1)} \in \mathcal{S}^+, \sigma^{(2)} \in \mathcal{S}^-$:
For $x \in \bigcap_{i=1}^n A_i^+$, we have $h_i^{(1)}(x) = \sigma^{(1)}(+\infty) = 0$, while for $x \notin \bigcap_{i=1}^n A_i^+$ have $h_i^{(1)}(x) < \sigma^{(1)}(+\infty) = 0$. Therefore

$$\lim_{\lambda \to +\infty} h^{(2)}(x) \begin{cases} \sigma^{(2)}(b) = \gamma, & \text{if } x \in \bigcap_{i=1}^n A_i^+, \\ \sigma^{(2)}(-\infty) = 0, & \text{otherwise} \end{cases}$$

where $\gamma$ can be any element of the image of $\sigma^{(2)}$, which is a non negative function. Therefore for $A^\cap = \bigcap_{i=1}^n A_i^+$

$$h^{(2)}(x) \approx \gamma \mathbb{1}_{A^\cap}(x).$$

If instead we assume $\sigma^{(1)} \in \mathcal{S}^-, \sigma^{(2)} \in \mathcal{S}^+$:
For $x \in \bigcap_{i=1}^n A_i^-$, we have $h_i^{(1)}(x) = \sigma^{(1)}(-\infty) = 0$, while for $x \in \bigcup_{i=1}^n A_i^-$ have $h_i^{(1)}(x) > 0$.

$$\lim_{\lambda \to +\infty} h^{(2)}(x) \begin{cases} \sigma^{(2)}(b) = -\gamma, & \text{if } x \notin \bigcup_{i=1}^n A_i^+, \\ \sigma^{(2)}(+\infty) = 0, & \text{otherwise} \end{cases}$$

where $-\gamma$ can be any element of the image of $\sigma^{(2)}$, that is now a non positive function. Therefore for $A^\cup = \bigcup_{i=1}^n A_i^+$

$$h^{(2)}(x) \approx -\gamma(1 - \mathbb{1}_{A^\cup}(x)) = \gamma \mathbb{1}_{A^\cup}(x) - \gamma$$

$\square$

For a more intuitive understanding of the class of functions that such constructed second layer can learn, in Figure 8, we report some samples from that class of functions.

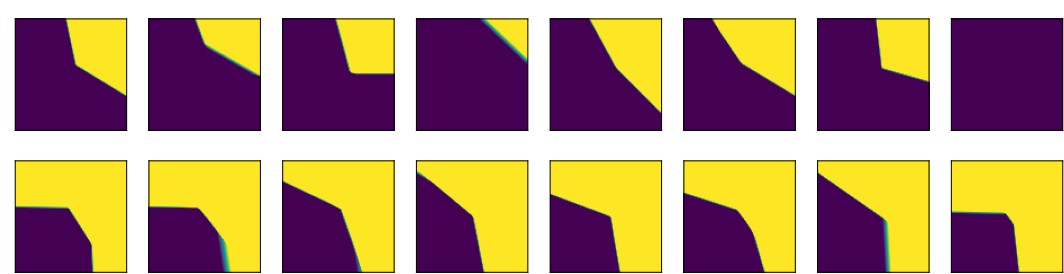

Figure 8: Examples of learnable indicator functions at the second hidden layer.

**Third layer construction** Finally, let us show that a hidden unit in the third layer can perform union and intersection operations when the second-layer representations are indicator functions of sets.

**Lemma 5.** *If $h_i^{(2)}(x) = \gamma \mathbb{1}_{A_i}$, there exists a rescaling factor $\delta \in \mathbb{R}_+$ such that a single unit in the third hidden layer of an MLP with non-negative weights, can approximate:*

$$h^{(3)}(x) \approx +\delta \mathbb{1}_A(x)$$

*for any $A = \bigcup_{i=1}^n A_i$ when $\sigma^{(3)} \in \mathcal{S}^+$, and for any $A = \bigcup_{i=1}^n A_i$ if $\sigma^{(3)} \in \mathcal{S}^-$*

We are finally ready to prove the main result.

*Proof of Theorem 1.* Since the function to interpolate is monotonic, for any couple of points $x_{i_1} < x_{i_2} : f(x_{i_1}) < f(x_{i_2})$ it is possible to find a hyperplane with non-negative normal, with positive and negative half spaces denoted by $A_{i_2/i_1}^+$ and $A_{i_2/i_1}^-$, such that $x_{i_1} \in A_{i_2/i_1}^-, x_{i_2} \in A_{i_2/i_1}^+$.

Let us now construct the sets:

$$A_{x_i}^\cap = \bigcap_{j:x_j < x_i} A_{i/j}^+ \tag{22}$$

$$A_{x_i}^\cup = \bigcup_{j:x_j > x_i} A_{i/j}^+ \tag{23}$$

This ensures that $x_j < x_i \implies x_j \notin A_{x_i}^\cap$. Also, since $A_{x_i}^\cap$ is obtained from the intersection of positive half-spaces, Lemma 4 ensures a hidden unit at the second hidden layer is able to learn $h^{(2)}(x) \approx \mathbb{1}_{A_{x_i}^\cap}(x)$. Now note that $A_{x_i} = \bigcup_{j:f(x_j) > f(x_i)} A_{x_j}^\cap$ contains only and all points $x_j$ such that $f(x_j) \geq f(x_i)$. Moreover, from Lemma 5, we know that hidden units in the third layer can approximate $\mathbb{1}_{A_{x_i}}$.

As per the previous layers, we show in Figure 9 some samples of functions that the third layer, constructed as just reported, can learn.

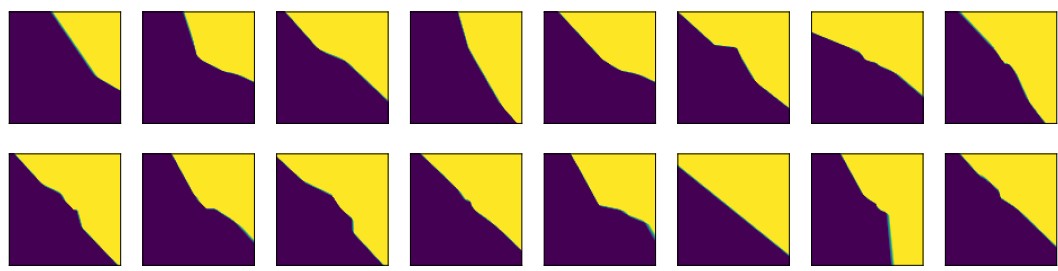

Figure 9: Examples of learnable functions at the third hidden layer.

**Fourth layer construction**  To conclude the proof, take the last layer parameters to be $w_i^{(4)} = f(x_{i+1}) - f(x_i), b^{(4)} = f(x_1)$. This produces the following function approximation

$$\bar{f}(x) = f(x_1) + \sum_i \mathbb{1}_{A_{x_i}} (f(x_{i+1}) - f(x_i))$$

. $\bar{f}(x)$ evaluated at any of the points $x_i$ provides a telescopic sum where all the terms elide, leaving $\bar{f}(x_i) = f(x_i)$. For the opposite activation pattern, the same result can be obtained in a similar fashion considering intersections of $A_{x_i}^\cup = \bigcup_{j:x_j > x_i} A_{i/j}^+$ instead. $\qquad\square$

