# OpenReview forum: "Unbounded Activations for Constrained Monotonic Neural Networks"
_ICLR.cc/2025/Conference — ICLR 2025 Conference Withdrawn Submission_

### Official Review · Reviewer_tpMq · 2024-10-18

**Soundness:** 4
**Presentation:** 2
**Contribution:** 4
**Rating:** 8
**Confidence:** 5

**Summary:**

This paper studies learning monotone functions with networks. The authors show a powerful expressivity result: non-negative weight networks with left- and right-saturated activation functions can approximate all monotonically increasing functions (thus all monotone functions via sign flip). They further show that non-positive weights lead to universal approximation of monotone functions even with convex activations such as ReLU, while these activations combined with non-negative weights can only approximate convex monotone functions. Overall, the authors present novel and impressive theoretical results.

This work further proposes an algorithm to utilize their theorems, achieving sufficient improvements on the tasks they studied.

**Strengths:**

The theoretical results are sufficiently impressive, and the proof techniques are clear and well-motivated. I found the proof easily understood. The authors also support their theorems with adequate numerical experiments.

**Weaknesses:**

The paper is barely written and not sufficiently revised. For example, numerous typos exist: Line 267, I suppose a $\gamma$ is missing before $1_A(x)$; Fig 2, caption looks messy; Line 473, a sentence is incomplete. I suggest all authors, especially those senior, to carefully revise the paper end to end since the current writing is a bit in fragments.

Improvements achieved in Table 1 looks marginal except for the heart disease dataset. However, I would not indicate negative opinions due to this as this is acceptable for this mostly theoretical paper. In contrast, I find this quite interesting because they were able to design a trick other than constraining the weights to enforce the monotonicity constraint.

**Questions:**

I have an additional question regarding the algorithm. The authors convert the constraint on networks weights to constraints on the prediction, i.e., design different forward pass for positive and negative weights. They simply claim this would mitigate the optimization difficulty because seemingly this will not lead to gradient vanishing. However, I did not find sufficient theoretical/experimental results supporting this other than the main results presented in Table 1. It would be better if the authors could clarify this.

---

> ### Author Response · Authors · 2024-11-16
> **Reply to Reviewer tpMq**
>
> Thank you for your thoughtful and detailed review. We are pleased that you find as much value in the theoretical results as we do. Indeed, our primary aim was to broaden the theoretical basis on constrained monotonic MLPs. We have carefully considered each of your comments and have made revisions accordingly. We highly thank you for the care and precision you took to report them. We are in the process of revising the manuscript to match the expected standard.
>
> **Replies to Weaknesses:**
>  1. Thank you for your detailed observations regarding the state of the manuscript. We acknowledge that the document requires a more thorough revision to meet the high standards expected. We have carefully re-examined the manuscript end-to-end, correcting the typographical errors, including the missing term in Line 267, and clarifying the caption of Figure 2. We have also addressed the incomplete sentence noted in Line 473. We appreciate your help in identifying these issues, and we are committed to ensuring the paper’s text is clear and professionally presented.
> 2. The primary benchmarks used in our study do not specifically test for the monotonic expression capabilities of networks, and their composition of monotonic and non-monotonic features can overshadow the importance of an expressive monotonic component. Our focus was not on optimizing these parameters extensively but rather on illustrating the applicability of our theoretical contributions in a fair comparative setup. Furthermore, it is possible that the benchmarks currently used are reaching a saturation point where significant improvements are increasingly difficult to achieve. We acknowledge this and, as mentioned in the conclusion of Section 3.3, plan to explore the development of new benchmarks that could more effectively test the capabilities of models designed under our theoretical framework. This feedback is invaluable, and we intend to further this aspect of our research to ensure that theoretical innovations are adequately tested for practical efficacy.
>
>  **Replies to Questions:**
>  1. Thank you for your question. Due to the page limit constraints, we were unable to delve into detailed theoretical or experimental analyses on every aspect of the model’s performance, including the specific optimization dynamics you mentioned. Our approach converts constraints on network weights into constraints on predictions by designing a forward pass that differs for positive and negative weights. This method aims to mitigate common optimization issues such as vanishing gradients and the problem of dead neurons. The use of non-bounded activations like ReLU helps alleviate the issue of vanishing gradients. Additionally, the risk of dead neurons, which is a significant concern in non-positive constrained MLPs, is notably reduced in our activation-switch architecture. This reduction is mainly due to the simultaneous use of both positive and negative weights. This provides a form of internal regularization and prevents all neurons from being inactive simultaneously, especially at initialization. While the main results presented in Table 1 focus on empirical performance, they also indirectly suggest that the network is effectively overcoming some traditional optimization barriers, as indicated by the relative success across various benchmarks. For future work, we plan to provide more detailed studies specifically addressing these optimization challenges better to substantiate the theoretical and practical benefits of our approach. More generally, we plan to do more analysis on the sensitivity and optimization properties of all proposed solutions in the literature for a more unbiased and comprehensive comparison.

---

> ### Comment · Reviewer_tpMq · 2024-11-16
>
> I want to thank the authors for the detailed reply.
>
> A small study is better than many words regarding the claim about vanishing gradients and other optimization problems. For example, as a minimal study, the authors could discuss (maybe plot empirically) the potential problems at initialization and how their approach solves this. Since their approach simply converts the input, the original argument from Kaiming's initialization should hold and maybe keep the variance roughly constant (but the variance after ReLU is changed due to the transformation, thus a closer look is required). Other arguments as presented by the authors could also help, but I encourage them to include a discussion and illustration supporting their claims about the optimization difficulty. Regarding page limit, such studies could be included in the appendix.
>
> I have carefully read the other reviews and found most complaints focus on the empirical algorithm. While I agree with many points raised by the other reviewers, I insist that this is not a huge problem for this mostly theoretical work. I have checked the full proof and did not find obvious mistakes in the steps. However, the authors should also include a detailed discussion about their empirical performance and highlight their main contribution more clearly in the main text.
>
> I maintain my original score.

---

> > ### Author Response · Authors · 2024-12-01
> >
> > Dear Reviewer,
> >
> > As the rebuttal process concludes, we wanted to sincerely thank you for your valuable suggestions and for recognizing the contributions of our work from the outset. Your feedback has been immensely helpful in refining our paper.
> > In response to your concerns, we have updated the manuscript and included an additional study on the optimization properties in the appendix.
> >
> > Thank you once again for your time and thoughtful review.
> > Best regards,
> > The Authors

---

### Official Review · Reviewer_TpwQ · 2024-11-02

**Soundness:** 3
**Presentation:** 3
**Contribution:** 3
**Rating:** 6
**Confidence:** 3

**Summary:**

This paper introduces a new method for building monotonic neural networks using unbounded activations like ReLU, avoiding traditional weight constraints. The proposed architecture dynamically adjusts activations to ensure monotonicity, simplifying optimization. Theoretical results show it can approximate any monotonic function, and experiments demonstrate improved accuracy on several datasets. This approach enhances the usability of monotonic networks in interpretable AI applications.

**Strengths:**

1. The paper introduces a new parametrization technique for monotonic MLPs, which removes the need for weight constraints and allows the network to adjust activations dynamically. This could improve flexibility and ease of optimization.

2. The authors provide a new theoretical result showing that MLPs with alternating unbounded activations are universal approximators for monotonic functions. This strengthens the theoretical foundation of monotonic neural networks.

3. By focusing on monotonic architectures, the paper addresses a key need in applications where interpretability is critical (e.g., fairness and transparency), making this work relevant for real-world deployment.

**Weaknesses:**

1. The experiments cover only a small number of datasets. Extending the evaluation to more diverse datasets and tasks would provide stronger evidence of the model’s generalizability and practical utility.

2. Although the authors propose a solution to avoid weight constraints, they do not fully address potential optimization challenges, such as sensitivity to initialization or convergence rates, which may impact the method’s robustness.

3. The paper’s theoretical section relies heavily on specific activation behaviors without enough real-world validation, which may limit the practical applicability of the theoretical claims.

**Questions:**

1. Could the authors clarify why ReLU and similar unbounded activations were chosen over traditional bounded activations in monotonic settings? What specific advantages do these activations bring in practical applications, beyond the theoretical universal approximation guarantee?

2. How does the proposed parameterization handle sensitivity to weight initialization? Did the authors test different initialization schemes, and if so, which were most effective? Further details here would be valuable for readers aiming to replicate the setup.

3. The experiments focus on a limited set of datasets. How does the model perform in other domains requiring monotonicity, such as environmental modeling or medical risk assessment? Expanding the range of tested applications could reinforce the paper's claims of generalizability.

4. The paper mentions that the new parameterization requires double matrix multiplications for weight splitting. Did the authors measure the computational overhead introduced by this approach? Reporting training times or memory usage compared to other methods would give insights into the scalability of this model.

---

> ### Author Response · Authors · 2024-11-17
> **Reply to Reviewer TpwQ**
>
> Thank you for your detailed review and the constructive comments provided. We are in the process of revising our manuscript to address the issues you’ve highlighted and to improve its overall presentation. We strive to make our responses as clear and thorough as the questions posed in your review. However, due to character limitations, we will break our reply into two parts to ensure a complete and transparent discussion. We believe in the value of the presented work and look forward to engaging in a constructive dialogue with you to refine it further and meet the standards of a publishable manuscript. Allow me to underscore a pivotal aspect of our research, which we consider crucial. Our paper aims to deepen the theoretical understanding of monotonic neural networks. Traditionally, bounded monotonic activations, such as sigmoid functions, have been fundamental in theoretical models within constrained MLPs. However, Theorem 1 of our paper broadens this perspective by showing that unbounded activations, which saturate on one side like the ReLU, are also viable. This advancement enables the application of most modern activation functions. Additionally, while it has been previously established that restricting weights to non-negative values with ReLU activations results in monotonic MLPs that do not universally approximate, our research introduces groundbreaking insights that using ReLU activations alongside non-positive weights does indeed permit the creation of monotonic MLPs that are universal approximators. Furthermore, we would like to highlight that we have been careful on the experimental side to match the community standard, testing our proposed solution in all aspects and in all the benchmarks that have been used in all previously published works. To conclude, we would like to highlight how the theoretical results obtained in Theorem 1 actually generalize the current state of theoretical results present in the literature, broadening the set of possible activations employable for monotonic MLPs to almost all activations used nowadays (consider Table 3 for a comprehensive list).
>
>  **Replies to Weaknesses:**
>  1. We appreciate your suggestion to expand the range of datasets used in our experiments. While we agree that a broader dataset pool would strengthen the evidence of our model’s generalizability, we chose these datasets because they represent the standard benchmarks commonly used in this field. All the published works cited in our paper that introduce new architectures utilize this exact set of datasets. This consistency allows for direct comparison of results and benchmarking within the established framework. However, we acknowledge the value of testing on a more diverse array of datasets and will consider this for future work to further validate and demonstrate the practical utility of our model.
> 2. Thanks to the newly introduced theoretical results, we introduced an architecture that leverages these new insights to propose an improved parametrization. Such parametrization addresses optimization problems such as gradient vanishing or dead neuron dynamics by construction. The former is addressed by the usage of ReLU like activations, and the second one by relaxing the weight constraint. Therefore, the proposed solution shares the same property of a normal MLP with ReLU activations, which is a widely studied and employed setting. Our experiments were designed to mirror those used in closely related works, like Constrained Monotonic NN, ensuring that our experiments follow the same framework, and thus the findings are comparable with previous works. Our contribution is intended to spur further research that can explore these practical challenges, building on the theoretical groundwork we have laid.
> 3. The focus of our paper is primarily on expanding the theoretical capabilities of constrained monotonic MLPs by broadening the set of applicable activation functions; therefore, it is a generalization of current theoretical results. This theoretical extension is significant as it moves beyond the traditional requirement of sigmoid-like activations, which were previously necessary to guarantee universal approximation properties. In the appendix, specifically in Table 3, we provide a comprehensive list of potential activation functions that can now be utilized, encompassing nearly all recently developed activations. In conclusion, the theoretical section, specifically Theorem 1, requires the activations only to be monotonic and to have one of the two sides approach C ∈ R in the limit: such constraints are satisfied by almost all real-world activations, as reported in Table 3. Therefore, the presented work actually relaxes the hypothesis that previously was necessary (i.e., the activations had to be monotonic and bounded).

---

> ### Author Response · Authors · 2024-11-17
>
> **Replies to Questions:**
>  1. The choice of ReLU and similar unbounded activations over traditional bounded activations such as sigmoid in our study is twofold: practical optimization advantages and theoretical flexibility. From a practical standpoint, ReLU-like activations have become predominant in most recent deep neural network architectures primarily due to their favorable optimization properties, which include avoiding issues like vanishing gradients that are more prevalent with sigmoid-like activations. These benefits are illustrated in Figure 4 in the appendix, where we compare optimization outcomes. From an empirical perspective, many groundbreaking works in AI use ReLU as activation, such as Transformers and ResNets, while theoretically, ReLU is often employed in the analysis of stability and representation abilities of networks. Therefore, the ability to use ReLU for monotonic MLPs allows for future theoretical analysis and potentially better empirical results. By demonstrating that unbounded, ReLU-like activations can also support universal approximation, our research expands the theoretical understanding and practical toolkit available for designing efficient and effective monotonic neural networks, making it more general.
>  2. We did not explore specialized initialization schemes in our experiments; instead, we utilized the default weight initialization provided by PyTorch. This ideally demonstrates that our proposed parameterization is robust to the choice of initialization, performing well without the need for any specific adjustments. For the sake of reproducibility and to aid others in replicating our setup, we have included the source code as part of the supplementary materials, but if you believe the manuscript would benefit from more technical details, we will proceed with adding them to the appendix.
>  3. We acknowledge the importance of testing our model across a variety of domains to enhance the claims of generalizability. For the current study, we deliberately chose to utilize the same set of datasets that are commonly employed in this research area, ensuring a direct and fair comparison with existing methods. This approach also helps mitigate discrepancies that might arise from variations in dataset-specific implementation details. It is worth noting that all the works cited in our paper, which comprehensively cover the recent literature, have also reported results on these benchmark datasets. This consistency across studies provides a reliable foundation for evaluating the comparative effectiveness of different approaches.
>  4. Thank you for your question regarding the computational aspects of our new parameterization. In our implementation, we utilized a naive approach involving double matrix multiplications, where the same matrix is used twice. This approach does not incur additional memory overhead due to reusing the matrix. While double matrix multiplication could potentially increase the computational load, the operation can be optimized significantly using custom matrix multiplication (mammal) kernels, which we plan to explore in future work. Currently, the naive implementation does not introduce significant overhead, as GPU operations are often more limited by data transfers rather than raw computation. Moreover, the activation-switch method used avoids the need for weight reparameterization (constraining weight to be non-negative), which can further complicate computational efficiency in other models compared to the proposed. The potential overhead from double matrix multiplications is comparable to using a larger single matrix with additional processing steps, which varies depending on network architecture and hardware capabilities. It’s important to note that the impact of these computational factors is highly dependent on the specific hardware used, the available computational budget, and the settings of the experiments. These variables can significantly influence performance metrics such as training times and memory usage. Our focus remains on expanding the theoretical understanding of constrained monotonic MLPs, with the practical implementation serving as a proof of concept demonstrating the viability of our theoretical advancements. In conclusion, we hope our responses have addressed all your concerns. We remain open to further collaboration to refine our paper, ensuring it reaches the publishability standards of the conference. We look forward to your continued feedback.

---

> > ### Comment · Reviewer_TpwQ · 2024-11-26
> >
> > Thank you for responding to the questions. Most concerns are resolved properly. I am willing to consider raising my score.

---

> > > ### Author Response · Authors · 2024-11-27
> > >
> > > Dear Reviewer,
> > > We are glad to hear that most of your concerns have been resolved and that you are willing to raise the score. Thank you very much for your response and for reconsidering the value of our work.
> > > It seems that the updated score is not yet reflected in the system. We would greatly appreciate it if you could check and confirm at your convenience.
> > > Please let us know if there’s anything further we can do or clarify regarding our work.
> > >
> > > Best regards
> > > The authors

---

> > > ### Author Response · Authors · 2024-12-01
> > >
> > > Dear reviewer,
> > >
> > > Given that we are entering the **final day of the discussion phase**, we just wanted to remind you that **the score has yet to be updated**.
> > > It would mean a lot to us if you could check to confirm.
> > > If, instead, you are still undecided on the matter, please let us know if we can help in clarifying any further concerns.
> > > We thank you for your time and efforts.
> > >
> > > Sincerely,
> > > the Authors

---

### Official Review · Reviewer_kcAu · 2024-11-03

**Soundness:** 3
**Presentation:** 3
**Contribution:** 3
**Rating:** 6
**Confidence:** 5

**Summary:**

This paper proposes a new neural network structure to reflect monotonic relationships between a specific features and the output in neural networks.  Traditional monotonic MLPs rely on non-negative constrained weights and saturating activation functions (e.g. Sigmoid, Tanh) to maintain monotonic relationships, which can cause optimization difficulties and limit expressiveness. The paper proves that by using alternating left-saturating and right-saturating monotonic activation functions, an MLP with either non-negative or non-positive constrained weights can serve as a universal approximator for monotonic functions. Additionally, the authors propose a new parameterization method (activation switch) that eliminates the need for weight constraints, thereby enhancing optimization stability and performance. Experimental results demonstrate that the proposed method achieves better approximation accuracy than existing monotonic neural network structures while preserving monotonicity and universal approximation properties.

**Strengths:**

$\bullet$ The proof is constructed with solid mathematical rigor.

$\bullet$ The proposed method (activation switch) is explained clearly and in an easy-to-understand manner.

$\bullet$ The experiments on real-world datasets for the proposed method were conducted appropriately.

**Weaknesses:**

Weakness 1. The categorization of related literature is unclear, and explanations are insufficient.
-  The explanations of related work in Section 1 (Introduction) and Section 2 (Related Work) are incomplete and would benefit from integration. (soft vs hard / CONSTRAINED MONOTONIC ARCHITECTURES vs HEURISTIC AND REGULARIZED APPROACHES)
-  There is an unclear expression on page 1, line 49-50. ("Such guarantees usually come at the cost of effectiveness.") Please explain the following sentence more clearly.

Weakness 2. Additionally, many relevant references are missing([1], [2]).

-   Both papers [1] and [2] are recent works in the field and should be included as comparison targets as related works. These studies should be cited in the paper.


Weakness 3. There are doubts about the contributions claimed by the authors, and most aspects raise concerns regarding novelty.

-  Despite the authors' efforts in their proof, it seems that the essential point is that a monotonic neural network requires positive weights (or an even repetition(e.g. $2$-layers, $4$-layers ... $2n$-layers) of negative weights) and a structure with activation functions that support monotonic increasing properties (including both convex and concave functions). I believe the proposed structure $activation switch$ in this paper does not deviate from this approach in the broader sense. If I missed anything, please explain more clearly how the proposed method differs from existing methods.

Weakness 4. The practical issue of gradient vanishing problem.
-  Concerns about gradient vanishing should be critically considered, especially given that monotonic neural networks can generally serve as universal approximators for arbitrary monotonic functions even with a shallow depth of 4. In fact, papers [2] and [3] report better performance than the proposed method and appear to have much simpler architectures, using lower depth network.

- As I understand it, using batch normalization along with existing methods seems to significantly alleviate the gradient vanishing problem. Is my understanding correct?



Weakness 4. The experimental results in this paper exclude several recent works in the field from comparative analysis. Although I do not believe that solely achieving SOTA performance defines the contribution of this study, there are concerns about cherry-picking in the reported results, or revisions may be necessary to better reflect the authors' claims in the experimental section.

- Paper [3] is mentioned in the main text but was excluded from the experiments. Paper [3] uses exactly the same datasets as the authors, and even shows better performance on some of them. To address concerns about cherry-picking, paper [3] should be added to the experiment.
- Paper [2] also constrains weights to be non-negative through re-parameterization (exponential transformation) and uses a saturated activation function. However, it achieves very good performance (SOTA) on several datasets. So, paper [2] also should be added to the experiment.

Weakness 5. (minor) There are multiple comma-separated phrases within single sentences, or sometimes incomplete sentences, making it difficult to understand.

- page1, line 71-74
- page3, line144-146
- page7, line 326-328
- page7, line 332-334
- page9, line 473 (incomplete sentence)

Weakness 6. It would be beneficial to include a discussion on the ethical considerations associated with using the COMPAS dataset. Given that this dataset involves sensitive information and has been widely discussed in terms of fairness and bias, it would strengthen the paper to address potential ethical concerns and how these were considered in your analysis.

References

[1] Yanagisawa, H., Miyaguchi, K., & Katsuki, T. (2022). Hierarchical lattice layer for partially monotone neural networks. Advances in Neural Information Processing Systems, 35, 11092-11103.

[2] Kim, H., & Lee, J. S. Scalable Monotonic Neural Networks. In The Twelfth International Conference on Learning Representations.

[3] Nolte, N., Kitouni, O., & Williams, M. (2022, September). Expressive monotonic neural networks. In The Eleventh International Conference on Learning Representations.

**Questions:**

1) If I understand correctly, since they use "point-symmetric activation functions" simultaneously, wouldn’t the authors' claim in "We prove that contrary ...  even convex ones like ReLU, is a universal approx"(at contribution section in Introduction) that it is "even convex once" be incorrect?

2) In the text, does "Constrained MLP" actually refer to "Constrained MNN"?

3) Is there any other reason why the experimental results from paper [3] were excluded?

4) It would be better to cite the officially published versions of papers, including paper [3], rather than the arXiv versions where possible.

5) Based on my understanding of the paper, it seems that having both "convex" and "concave" types of activation functions within the network is a more important key point than simply having "left-saturating" and "right-saturating" activations. Could you explain the difference between these two concepts in more detail?

(minor)
1) page9 line 473 "The only requirement for this method to work, is to have the input features to be", Isn't it an incomplete sentence?

**Details Of Ethics Concerns:**

The paper uses the COMPAS dataset, which has also been utilized in related studies that serve as benchmarks in this field. Given that research on monotonicity often overlaps with fairness studies, the use of this dataset can be justified. However, it is essential to address the ethical concerns associated with the COMPAS dataset. This dataset has been widely criticized for potential biases and fairness issues, as it includes sensitive data and has been shown to sometimes produce biased outcomes against specific demographic groups. Including a discussion on these ethical issues, as well as the measures taken to mitigate potential bias in the analysis, would strengthen the paper’s ethical rigor and transparency.

---

> ### Author Response · Authors · 2024-11-17
> **Reply to Reviewer kcAu**
>
> Thank you for your detailed review and insightful feedback. We greatly appreciate your expert input and are dedicated to improving our manuscript according to your recommendations. Our goal is to address your review with clarity and depth. To ensure a full and transparent response while accommodating character limits, we will present our reply in two separate sections.
>
> We understand the necessity of presenting our work in the most robust and comprehensible manner, and we hope to work closely with you to enhance our study to meet the publication standards. Currently, we are revising the manuscript to address the concerns you have raised. Concurrently, we seek to engage in a constructive dialogue with you to clarify and address any weaknesses identified. We believe that this collaborative effort will significantly enhance the quality of our work and ensure it meets the high standards expected at the conference.
>
> We wish to emphasize a critical aspect of our research, which aims to provide a theoretical contribution to the field of monotonic neural networks over possible improvements in metrics over the state-of-the-art. Traditionally, theoretical models for these networks have relied on bounded monotonic activations, such as sigmoid functions, within constrained MLPs. However, our Theorem 1 expands upon this by demonstrating that unbounded monotonic activations that saturate on one side, like ReLU, are also viable. This advancement enables the use of nearly all contemporary activation functions. Building on these findings, we prove that a negatively constrained ReLU MLP is a universal approximator, contrary to its counterpart with positively constrained weights, which has been shown not to be a universal approximator.
>
> **Replies to Questions:**
>  1. That’s a good observation, but it appears there might be a misunderstanding regarding the activation functions used in our model. That contribution is about non-positive constrained MLPs (with weights all negative or zeros). In this case, we are not using point-reflected activations simultaneously, but only a single one (see proposition 5 for more detail). The core of this contribution is pointing out that there is a very clean and simple architecture that satisfies the ”alternating” saturation requirement, i.e., fixing the activation and using negative weights.
> 2. No, ”Constrained MLP” refers generally to MLPs with constrained weights and monotonic activations. ”Constrained MNN” is a specific subset within this broader category.
> 3. The exclusion of paper [3] from our experimental comparison was deliberate due to its fundamentally different approach to addressing monotonicity. Additionally, [3] does not provide a broad comparative analysis with other works, which could affect the fairness and accuracy of direct comparisons. We focused on results from Constrained MNN where we could ensure compatibility in datasets and experimental settings for a more reliable comparison. These factors—differing approaches and the absence of a comprehensive benchmarking in [3]-were our primary reasons for not including it in our comparison.
> 4. Thanks. We will update our citations to reference the officially published versions of the papers.
> 5. Indeed, the distinction between ”left-saturating” and ”right-saturating” activations and ”convex” and ”concave” activations is crucial and might be misunderstood. Left-saturating and right-saturating refer to the behavior of the activation function as the input tends towards negative or positive infinity, respectively. For example, sigmoid and tanh are both left and right saturating as they approach constant values at the extremes, but they are neither purely convex nor concave across their entire domains. Conversely, convex and concave functions refer specifically to the curvature of the activation function. For instance, a leaky-ReLU is generally convex due to its linear increase. Still, it does not satisfy the saturating property on both sides, which is necessary for ensuring the network is a universal approximator in constrained architectures. One possible example is $f(x) = max(0,sin(x) + x)$.
> 6. Yes, thank you for pointing it out. It appears that the sentence was inadvertently left incomplete.
>
> In conclusion, we sincerely appreciate the opportunity to improve our manuscript based on your detailed feedback. We are dedicated to making necessary revisions that not only address your concerns but also enhance the clarity, depth, and impact of our work. We hope that through this collaborative effort, our manuscript will be brought to a publishable standard that meets your expectations and contributes meaningfully to the field. We look forward to your continued guidance and feedback in this process.

---

> ### Author Response · Authors · 2024-11-17
>
> **Replies to Weaknesses:**
>  1. We acknowledge that the categorization and explanations of related literature in Sections 1 and 2 could be clearer and more integrated. We will revise these sections to provide a more structured overview of the related works, distinguishing more clearly between constrained monotonic architectures and heuristic and regularized approaches. Regarding the sentence on page 1, line 49-50, we agree that it could be expressed more clearly. The phrase ”Such guarantees usually come at the cost of effectiveness” refers to the trade-offs involved when enforcing hard-monotonicity in neural networks. Empirically, such constraints can complicate the optimization process, leading to issues like vanishing gradients with sigmoid-like activations or dead neurons with ReLU6 activations. We will clarify this explanation in the revised manuscript.
> 2. Thanks. We aimed to have a comprehensive literature review, but we overlooked these recent works. We will proceed with adding them.
> 3. We want to clarify the novelty of our work. Prior literature on monotonic MLPs employed activations that approximate the Heaviside function (sigmoid or tanh) to prove universal approximation. Our contribution significantly diverges from these methods by demonstrating that a combination of left-saturating and right-saturating activations is sufficient to construct universal approximators. This is a generalization of previous theoretical results and enlarges the pool of possible architecture and activations. Notably, thanks to such findings, we provide evidence that MLPs with only negative weights, an even-number of layers and ReLU-like activations are universal approximators. This contrasts sharply with similar architectures that use only positive weights, which are provably not universal approximators.
> 4. Thank you for your comments. Let us clarify the distinctions between our approach and those mentioned in papers [2] and [3].The architecture described in [3] indeed focuses on maintaining the Lipschitz constant of the MLP, employing weights normalization to achieve this. This method, while effective, imposes complex constraints on the architecture and selection of possible activations, which can be a significant limitation (as highlighted by the authors themselves). Our proposed architecture in contrast, leverages a broader range of activation functions and offers a simpler yet effective parametrization. Regarding paper [2], while it does propose a potentially more scalable architecture, it lacks the theoretical underpinnings that ensure the network remains universally approximative under its constraints. Our work complements and could be integrated in such approaches, providing theoretical backing and enhancing their practical applications. As for the use of batch normalization in constrained monotonic MLPs, we suggested this as a novel potential solution to mitigate initialization issues. To our knowledge, this is a novel proposal and has not be extensively tested it in this specific context. Importantly, batch normalization might not sufficiently address the vanishing gradient problem during training (for large γ). In the appendix, we provide a simple example (Figure 4) illustrating how constrained architectures exacerbate the vanishing gradient problem of sigmoid-like activations, underscoring the challenges and nuances of addressing this issue.
> 5. Since our goal in the experimental comparison is mainly to validate our theoretical results, we simply reported results from Constrained MNN since we could easily reproduce the experimental setup, ensuring a fair comparison. You are correct that while paper [3] was mentioned, it was not included in the comparative analysis. This is because it would have been difficult to ensure that the reported results were obtained following the same dataset and preparation. Since it tackles the monotonicity from a completely different perspective, we believe that the exclusion is warranted. Regarding paper [2], which achieves SotA performance, we only now acknowledge its existence and we agree that contrary to [3], it works in the same area as our work and shares a comprehensive comparison. It seems that they are trying to address a different issue, in particular, the scaling issue of monotonic MLPs. Our contributions are orthogonal to theirs and could probably be easily be merged. However, at this stage, new results regarding our methodologies would be prohibitive, and we thus believe it’s better to stick with the work already presented. If you still find this work relevant for the comparison, we will proceed to consider it also for the final comparison, in addition to the discussion in related work from point (2). We want to highlight that we highly appreciate your acknowledgment that achieving SotA results is not the sole measure of a study’s value.
> 6. Thanks, you are correct. We will revise the manuscript and address them.
> 7. Thanks. We agree, and we will add a dedicated section.

---

> > ### Comment · Reviewer_kcAu · 2024-11-21
> >
> > Thank you for your detailed rebuttal and responses.
> >
> > I appreciate the authors' efforts and the willingness to improve this work. However, considering the high standards of the conference, I believe there are still some critical areas that require further refinement.
> >
> >
> >
> > 1) Concerns Regarding Theoretical Contributions
> >
> > I understand that the main contribution of the proposed method is to demonstrate that a universal approximator for monotonic functions can be achieved without bounded activation functions, using only ReLU and ReLU'. However, I still have some concerns regarding Definition 1 and Theorem 1.
> >
> > It appears that the authors assume ReLU $=max(0,x)$  belongs to $S^-$ and ReLU' $=-max(0,-x)$ belongs to $S^+$ when conducting their proofs and experiments. However, based on the given definitions, not only does ReLU' belong to  $S^+$, but so does $max(0,-x) \in S^+$. Under such conditions, the composition of convex functions remains convex, which means that the proposed method might fail to be a universal approximator for arbitrary monotonic functions (e. g. $x+sin(x)$), as claimed in Theorem 1.
> >
> > This aligns with my earlier remarks in the weakness 3 and question 5, where I pointed out that including both convex and concave hinges is critical for a universal approximator for monotonic functions.
> >
> > If my observation is correct, a significant revision to the claims and proofs in the paper might be necessary.
> >
> >
> >
> > 2. Concerns Regarding the Fairness of Experimental Results
> >
> > I appreciate the authors' explanation and their efforts to ensure fair comparisons in the experiments. However, I believe greater caution is needed when discussing the exclusion of benchmark results, especially when justifying why specific results were omitted. For example, the authors state:
> >
> > “Additionally, [3] does not provide a broad comparative analysis with other works, which could affect the fairness and accuracy of direct comparisons.”
> >
> > I am not fully convinced that [3] (LMN) lacks broad comparative analysis with other works. Upon reviewing the paper, it appears that LMN conducts proper comparisons with other methods using five benchmark datasets. This also applies to [2] (SMNN), whose experimental results on the same datasets should be included in Table 1.
> >
> > As I mentioned earlier, achieving SOTA is not the sole measure of a study's value, and I fully agree with this perspective. However, I strongly recommend that the authors include the results from [2] and [3] in their tables, as doing so will enhance the fairness and credibility of the presented experimental results.

---

> > > ### Author Response · Authors · 2024-11-21
> > >
> > > Thank you for your response. We sincerely appreciate the time and effort you’ve dedicated to reviewing our work. We are especially grateful for the level of detail and care you’ve demonstrated in your reviews.
> > >
> > >
> > > **Concerns on soundness of theoretical results**
> > >
> > > Our theoretical result holds for monotonically non-decreasing activations, reason why in Table 3 some parametrized activation require special care in order to be used. Due to such limitation $\max(0,-x)$ is not a qualified activation. Indeed, in Theorem 1, we require _"provided that the activation functions are monotonic and alternate saturation sides"_. However, by "monotonic" we refer to "monotonically non-decreasing", as reported in footnote 1. We recognize that the main theorem of a paper should not rely on a footnote to be correctly interpreted. In addition, the final part, _"That is, either of the following holds:_ $\sigma^{(1)} \in \mathcal{S}^-, \sigma^{(2)} \in \mathcal{S}^+, \sigma^{(3)} \in \mathcal{S}^-,|   \sigma^{(1)} \in \mathcal{S}^+, \sigma^{(2)} \in \mathcal{S}^-, \sigma^{(3)} \in \mathcal{S}^+$", makes it seem like alternating saturation is the only assumption needed.
> > > Given these observations, we have revised the theorem statement to make it more precise and clear. Thanks for pointing out this issue, we believe that this revision is highly beneficial and hopefully addresses your concerns.
> > >
> > > **Convexity and saturation**
> > >
> > > We agree that MLPs with non-negatively constrained weights and convex[concave] activations cannot be universal approximators. Indeed, the requirement of alternating left-saturating/right-saturating monotonic non-decreasing activations, implies that not all activations can be convex[concave].
> > > However, our result does not build on the activation convexity/concavity property, but only on the saturation, which is paramount to build the level-sets in our construction. For example, the presented activation in [1], JumpReLU, is a valid activation for our theorem, yet is neither convex nor concave.
> > >
> > >
> > > **Concerns Regarding Experimental Results**
> > >
> > >
> > > Regarding LMNN, the authors compare the presented method with a very restrictive pool of alternatives (with only a single other alternative method per dataset). The fact that they only compared to a single competitor, and that such result do not exactly match the one obtained with our setup, highlights that the setup used in LMN differs from our. For this reason, we believe that reporting their stated performances would undermine the fairness of the final comparison.
> > > On the other hand, SMNN compares their approach to the considered one in an extensive manner. Even though we are not certain that the setup is the same, the results reported are extremely similar to the one found in our setup. Thus, Upon further reflection, we agree that the inclusion of SMNN results would provide a more comprehensive comparison. For this reason, in the final version of the paper, we will include the results of SMNN in Table 1.
> > > Since the main aim of the paper was broadening the theoretical basis of monotonic constrained MLPs, the final comparison simply wanted to show that by using _only_ the new results, it is possible to build new architectures that are competitive with SoTA. For this reason, we felt that the fairness of the comparison was more important then comprehensiveness. Although this is not our primary focus, we would like to highlight that the proposed method still achieves comparable results to the SoTA, even when evaluated against a broader set of competitors.
> > > If, after all of this observations, you still find the inclusion of LMNN crucial, we are willing to do so, since we share the perspective that the final empirical comparison should be as fair and comprehensive as possible.
> > >
> > >
> > > [1] Rajamanoharan, Senthooran, et al. "Jumping ahead: Improving reconstruction fidelity with JumpReLU sparse autoencoders." arXiv preprint arXiv:2407.14435 (2024).

---

> > > > ### Comment · Reviewer_kcAu · 2024-11-24
> > > >
> > > > Thank you to the authors for their diligent and detailed rebuttal. Throughout the review process, I was impressed by the authors' efforts to improve their work, which has significantly enhanced the quality of the paper.
> > > >
> > > > Furthermore, most of my concerns have now been fully addressed.
> > > >
> > > > 1. Comments on Theoretical Contributions
> > > >
> > > > The questions I raised regarding $\textbf{Definition 1}$ and $\textbf{Theorem 1}$ were thoroughly resolved through the authors' rebuttal. I now fully acknowledge the main contribution of this paper, which is the demonstration of the universal approximator property without relying on heavy-side functions. This is indeed a clear and significant contribution of this study.
> > > >
> > > >
> > > > 2. Comments on the Fairness of Experimental Results
> > > >
> > > > Upon reviewing the revised manuscript, I confirmed that the experimental results of Scalable MNN [2] have been properly included. This has significantly improved the quality of the paper, particularly the validity and fairness of the experimental results. Regarding LMNN [3], while I personally believe that including its results would be beneficial, I respect the authors' position on this matter.
> > > >
> > > >
> > > > As a result, I believe that this constructive discussion during the rebuttal period has significantly improved the quality of the paper. I am revising my score from $\textbf{3 to 6}$. The reason I cannot assign a higher score, despite the theoretical contributions of the paper, is that the empirical comparisons still fall short in my view. Specifically, the methods compared in the experiments exhibited issues like gradient vanishing or dead neurons, which I believe did not sufficiently support the theoretical contributions with empirical evidence.
> > > >
> > > > As discussed during the rebuttal period, this paper has its own unique contributions beyond merely achieving SOTA results. I sincerely thank the authors for their thoughtful and dedicated rebuttal.

---

> > > > > ### Author Response · Authors · 2024-11-24
> > > > >
> > > > > We want to begin by sincerely thanking you for your detailed and thoughtful engagement throughout the review process. Your constructive and understanding approach has significantly improved the clarity and rigor of this work, elevating its overall quality to a level we are genuinely proud of. The current version of the paper owes much of its strength to your insightful feedback, and we are deeply grateful for the time and effort you dedicated to helping us refine our contributions.
> > > > >
> > > > > We fully understand your reasoning for not assigning a higher score, and we respect your perspective. In the appendix (Figure 5&6), we added a small study on vanishing gradient. It can be seen that sigmoidal constrained MLPs exhibit vanishing gradients, and the same experiment run with ReLU6 leads to _all_ dead neurons ($\frac{df_\theta(x)}{d\theta}=0$). We recognize that this aspect necessitates a more extensive study, and we plan to explore it in more details in the future. Nonetheless, we are truly thankful for your thoughtful reconsideration of our work and for recognizing its unique contributions.
> > > > >
> > > > > Thank you for your time, effort, and support throughout this process.

---

### Official Review · Reviewer_FcMS · 2024-11-05

**Soundness:** 3
**Presentation:** 3
**Contribution:** 2
**Rating:** 3
**Confidence:** 4

**Summary:**

The paper is about constructing monotonic MLPs with unbounded activation functions, to ease the optimization and mitigate the issue of saturation when bounded activation functions are used in prior works. The proposed construction involves an activation function whose direction is dynamically determined depending on the sign of the weight in the linear layer.

**Strengths:**

* The paper proposes a novel construction for monotonic MLPs with unbounded activation functions by introducing an activation function whose direction is dynamically determined depending on the sign of the weight in the linear layer.
* The paper theoretically proves that the constructed MLPs are universal approximators for monotonic functions.

**Weaknesses:**

*  The goal of the paper is on addressing optimization issues, but empirical gains are very limited. There is no gain on three out of the five experimented datasets. On the other two datasets, although the mean of the results is improved, the variance is also much larger. Overall, the empirical improvement is quite marginal, which makes the paper not convincing enough.
* The experiment section is also very short and the authors have not investigated the results further. For example, there is no result showing whether the optimization has been eased (with loss curves, scaling behavior, etc.)

Thus, on the empirical side, I think the paper is still quite preliminary and not yet ready for publication.

## Final response

Since I can no longer post additional messages, I am posting my final response to the authors here. My position regarding this paper remains the same, [as explained in my last response](https://openreview.net/forum?id=N1DKrLIKhT&noteId=ncbNUcaRZ4).

Now I will respond to the authors' [latest response](https://openreview.net/forum?id=N1DKrLIKhT&noteId=L5d3OYv6mS).

First of all, the authors inappropriately blamed that “the reviewer decided to wait until literally the last few hours to post his response”. The authors inappropriately assumed the pronouns of the reviewer. It makes no sense to say that I “decided to wait”. Note that I have responded to the initial rebuttal two weeks ago. My recommendation on this paper has been consistent, from my initial review, to my initial response to the rebuttal, and then my follow-up responses, and the concerns from my initial review have not been addressed by the rebuttal or follow-up responses which have been making misleading claims.

**I want to emphasize again that I don’t think the theory in this paper is a more general one and the authors are making misleading claims.** Under the same constraint with non-negative weights, the authors removed a restriction (the activation function must saturate on both sides and now an activation which only saturates on one side can be used), but the authors also introduced a new restriction (alternating saturation sides are required, which means that we have to use alternate activation functions every two layers, instead of the same activation function for all the layers). This is thus not a generalization. It has been reflected [by the authors themselves](https://openreview.net/forum?id=N1DKrLIKhT&noteId=RYasdvC09j):

>Until now, constrained monotonic MLPs were shown to be universal approximators only using the threshold activation (sigmoid-like), thus with both sides saturating. Instead, we show that **you only need activations that alternate sides of saturation**.

The authors then said:

>it is indeed possible to use the same activation function by ensuring all weights are negative

Note that this has changed the settings (from non-negative weights to non-positive weights). Although you can use the same activation function now, another new thing has been introduced (non-positive weights). The authors again remove a restriction but add another new restriction. This is again not a generalization.

On the empirical side, the paper only has very preliminary results. E.g., Figure 4 in the appendix showed some training loss curves on a toy setting, without further investigation on real datasets. It is also unclear if the difference in the training curves can be caused by the need of different hyperparameters when the models are different (e.g., possibly one setting requires a larger learning rate and then the loss may descend faster).

The paper requires much additional work to significantly improve the experiments which are still very preliminary for now. The experiments are necessary in order to justify that the theoretical analysis on model settings newly introduced by the authors (either alternating saturation sides or changing the sign of weights) are useful.

**Questions:**

Why is the empirical improvement so marginal, despite the theoretical insights shown in the paper?

---

> ### Author Response · Authors · 2024-11-16
> **Reply to Reviewer FcMS**
>
> Thank you for your review and the insights provided. We would like to briefly highlight a key aspect of our work, which we believe is critically important. This paper primarily seeks to contribute a theoretical foundation to the community of monotonic neural networks. Historically, theoretical proofs have necessitated the use of bounded monotonic activations (similar to sigmoid functions) in constrained MLPs. However, Theorem 1 in our study extends this understanding by demonstrating that unbounded activations, as long as they saturate on one side (such as ReLU), can also be effectively utilized. This finding allows for the use of nearly all contemporary activation functions.
> Moreover, while previous research has shown that constraining weights to be non-negative and employing ReLU activations results in MLPs that are monotonic but not universal approximators, our novel theoretical insights reveal that using ReLU activations with non-positive weights enables the creation of monotonic MLPs that are, indeed, universal approximators.
>
>  **Replies to Weaknesses:**
> 1. The contributions of the paper are mainly theoretical. In previous works, positively constrained MLPs with convex monotonic activations (ReLU) were shown not to be universal approximators; we show instead that it is possible to construct universal approximators using a much broader class of activations by simply using negatively constrained MLPs. In fact, the main contribution of the paper, (Theorem 1) is even more general, showing that the only requirements is the presence of a left-saturating activation and a right-saturating activation (ReLU-like), contrary to previous work that required bounded activations (sigmoid-like). Such theoretical results are general and not tight to the proposed formulation (activation switch), which shows one possible way of exploiting the new theoretical findings without the need to manually set the saturating directions. The experimental evaluation is meant to confirm that the theoretical results are also backed by empirical performance competitive with State-Of-The-Art.
> 2. Thank you for your insightful comments. We appreciate the opportunity to clarify the contributions and intentions of our work. The primary contributions of this paper are indeed theoretical. In contrast to prior studies that demonstrated that positively constrained MLPs with convex monotonic activations like ReLU are not universal approximators, our paper proves that by employing negatively constrained MLPs, one can achieve universal approximation abilities with a much broader class of activations. This theoretical advancement is encapsulated in Theorem 1, which also establishes that the mere presence of one left-saturating and one right-saturating activation function (akin to ReLU) suffices for universal approximation abilities, diverging from previous assumptions requiring bounded activations (similar to sigmoid functions). It’s important to note that these theoretical insights are broad and not strictly tied to the specific activation switch proposed in our experiments. This activation switch serves as an illustrative example of how our theoretical findings can be practically applied without manually setting the saturating directions. While our experimental results show marginal improvements and variability across different datasets, they are primarily intended to demonstrate that our theoretical results can be empirically viable and competitive with current state-of-the-art techniques. Our aim was to extend the theoretical understanding of constrained monotonic MLPs significantly, rather than focusing primarily on architectural innovations. We believe these theoretical advancements provide valuable groundwork for future research in this area, offering new directions for exploring and designing neural network architectures. We hope that this clarification underscores the significance and utility of our theoretical contributions.
>
>  **Replies to Questions:**
>  1. The architecture we investigated is an extension of the existing ”Constrained Monotonic NN” model, which limits the potential for dramatic empirical gains. Our primary contribution, detailed in Theorem 1, eliminates the need for a third class of activations, streamlining the architecture and reducing the need for manual tuning without compromising performance. This theoretical advancement simplifies the model significantly and is designed to maintain competitive performance rather than achieving substantial empirical improvements. The marginal empirical results reflect this focus on theoretical robustness and architectural simplicity. We anticipate that the benefits of our approach will become more evident as it is applied in more diverse contexts. In closing, we appreciate your guidance and believe that through this dialogue, we can achieve a version of the manuscript that you will also find suitable for publication. We look forward to your continued feedback.

---

> ### Comment · Reviewer_FcMS · 2024-11-18
> **Satisfactory empirical results are necessary**
>
> Both the authors and Reviewer tpMq are using an argument that the limited experiments with limited empirical improvement in this paper is not important because this work is “mostly theoretical”. I don’t find this argument reasonable here.
>
> In fact, this paper just reads like a regular ML paper proposing a new method, with a theoretical motivation and analysis, followed by empirical results. I don’t think it can be considered as mostly theoretical, as the paper is actually proposing a new model (just like other regular ML papers) instead of theoretically analyzing an existing technique. In this case, I believe a consistency between theoretical results and empirical results is necessary, and the empirical results should be able to demonstrate the advantage of the new model proposed.
>
> The paper is claiming that the proposed construction can be used for “enhancing optimization stability and performance” and “making the optimization more stable and less sensitive to initialization”, yet the experiments have not been able to sufficiently demonstrate the benefits. The paper also claims to achieve SOTA as “we show that we can achieve state-of-the-art performances”, which is not true.

---

> > ### Author Response · Authors · 2024-11-20
> >
> > Thanks for the clarification regarding your review. However, we feel that our theoretical and empirical results might have been slightly miss-interpreted.
> >
> > **Generality of theoretical results.**
> >
> > Until now, constrained monotonic MLPs were shown to be universal approximators only using the threshold activation (sigmoid-like), thus with both sides saturating. Instead, we show that you only need activations that alternate sides of saturation. This finding generalizes previous results, allowing for the usage of unbounded activations.
> > Only thanks to this new theoretical result, we are now able show that an MLP with negative weights and ReLU activations is a universal approximator, contrary to its counterpart with positive weights, which is not.
> > Furthermore, only thanks to these results, you can show that in “Constrained Monotonic Neural Network”, the bounded activation is actually unnecessary (at least from a theoretical standpoint).
> > In addition, citing the original authors of “Constrained Monotonic Neural Networks”, _“The best results in our experiments were achieved by neural networks with a significantly smaller number of layers [compared to O(d) requirement].“_. Our theoretical results give a theoretical explanation for this observation.
> >
> > **Significance of empirical results**
> >
> > However, we felt that a theoretical result is much more useful if applicable, and thus that can be used to build new approaches.
> > For this reason, we developed an architecture that only relies on the new theoretical findings (thus that could not have been shown to be universal approximator up until now), that nonetheless achieves SoTA results. We do agree that the improvement is marginal, but nonetheless matches almost always or improves over the SoTA. Indeed, the proposed approach does not differ in architecture significantly to Constrained MNN, thus major improvements were unlikely.
> > Still, the objective of this paper was not to propose a novel SoTA architecture. This is the main reason behind the fact that little to no hyperparameter tuning was performed on the architecture, compared to the main competitor (Constrained MNN) which instead applied Bayesian Optimization to optimally find their results. Nonetheless, the empirical results are still on par.
> > Generally speaking, this shows that these new theoretical tools allows the construction of effective monotonic MLPs. However, we agree that the SoTA performance stated in the paper might be bold, thus we have revised it in the new version we just uploaded, with all the fixes requested by all the reviewers. Furthermore, this new version includes in the appendix a small study on stability of monotonic MLP, showcasing the vanishing/exploding gradient problem in constrained MLPs, in addition to the ones already presented in Appendix A.4.

---

> ### Author Response · Authors · 2024-12-01
>
> Dear reviewer,
>
> Given that we are entering the last day of the rebuttal process, we would like to kindly ask you if our previous comment addressed your concerns. If this is not the case, the following is a brief summary of the changes we made in the revised manuscript that directly relate to your concerns:
>  - The main focus of the paper is Theorem 1, not the proposed architecture. The revised version makes this more explicit.
>  - We have added a small-scale study on optimization proprieties (dead neurons and vanishing gradients).
>  - We have included Scalable MNN in the comparison to enrich the empirical results.
>
> Also, please consider that the proposed architecture is a structural simplification of Constrained MNN, motivated by our theoretical results. Hence, the empirical improvements, even if marginal, are confirmation that the insights provided by Theorem 1 can be used to improve/simplify existing architectures.
> For a more detailed discussion please refer to our previous responses and to the general comment.
> If you do not find our response satisfactory, please let us know what we can do to clarify further.
> We thank you for your time and effort in helping us improve our work.
>
> Sincerely,
> the Authors

---

> ### Comment · Reviewer_FcMS · 2024-12-03
> **Reply**
>
> I don’t think it convincing to say that this “generalizes” previous results. **I believe the theoretical contributions have been overclaimed (and overestimated by some other reviewers).** In previous results, you can use the same activation function in every layer. However, in the new results here, you have to “alternate sides of saturation” and thus essentially use alternate activation functions for every two layers. **This seems to remove the restriction on the activation being “bounded”, but it adds a new restriction -- you can’t use the same activation function for all the layers any more.** This is not a generalization and not a “structural simplification”. The new change also makes the architecture different compared to regular NNs, yet such a change is not justified (given that there is no empirical improvement).
>
> Therefore, the paper is actually proposing some new architecture (with activations that alternate sides of saturation every two layers) and proving that this construction is a universal approximator. However, the paper has not been able to demonstrate that the new construction is empirically useful as I mentioned earlier, and thus it has not demonstrated that a theoretical analysis alone on this particular architecture (which is not more general than existing ones) is a significant contribution.
>
> As I said earlier, the claim that the focus of this paper is on a theoretical analysis is quite unconvincing and misleading. The theoretical analysis is not done for existing models which have been demonstrated to work well by previous works. If a theoretical paper analyzes some existing model (which previous works have demonstrated that the model is meaningful) and provides significant theoretical insights, it would be a significant theoretical contribution. However, here the theoretical analysis is done for something newly proposed in this paper, yet the paper has not been able to demonstrate that the new thing (which eliminates a restriction but adds a new restriction) is meaningful through experiments. Therefore, I believe a theoretical analysis alone is not enough to claim a major contribution.
>
> For experiments, I didn't mean that you must achieve SOTA. Instead, empirical results should justify claims made in the paper regarding the newly proposed architecture (“enhancing optimization stability and performance” and “making the optimization more stable and less sensitive to initialization”). This doesn't have to be SOTA, but experiments have to justify that the new method at least has sufficient benefits.

---

> ### Author Response · Authors · 2024-12-03
>
> It seems the reviewer has severely misunderstood many core aspects of this work. Regrettably, **the reviewer decided to wait until literally the last few hours to post the response**, leaving no possibility for further discussion.
>
> Here are the reasons why the raised concerns are inaccurate/wrong:
>  1. > it convincing to say that this “generalizes” previous results
>
> our contribution is a strict generalization of previous ones [1]. Indeed, using **the same bounded activation across all layers is a subcase of Theorem 1**.
>
>  2. >In previous results, you can use the same activation function in every layer. However, in the new results here, you have to “alternate sides of saturation”
>
> it is indeed possible to use **the same activation function** by ensuring all weights are negative, as clearly demonstrated in Section 3.3.
>
>  3. >The new change also makes the architecture different compared to regular NNs, yet such a change is not justified
>
> the proposed architecture **is not different from regular constrained NN**: it is a compact and efficient representation of a NN twice the size with block diagonal $W$ matrices.
>
>  4. >The theoretical analysis is not done for existing models which have been demonstrated to work well by previous works
>
> as previously stated and also present in the paper, **Theorem 1 can be directly applied to [2] to simplify the architecture (dropping the saturating activation) and to explain their empirical results**.
>
>  5. >the theoretical analysis is done for something newly proposed in this paper, yet the paper has not been able to demonstrate that the new thing (which eliminates a restriction but adds a new restriction) is meaningful through experiments
>
> **boundedness means 2 requirements, left and right saturation. Our results use ONLY either left or right saturation**. This clearly shows cases that we are not adding any more requirements, and as stated before, we are strictly generalizing previous results (which, again, implies no newly introduced restriction).
>
>  6. >empirical results should justify claims made in the paper regarding the newly proposed architecture (“enhancing optimization stability and performance” and “making the optimization more stable and less sensitive to initialization”)
>
> as stated before, **in Appendix A.4 there indeed is a study showcasing that**, which seem to have been ignored by the reviewer. Furthermore, as stated multiple times, the empirical results are meant to showcase that using **ONLY** the new theoretical findings you can build new Monotonic NN which are as expressive as SoTA.
>
> We believe that our work has been severely misunderstood, and we hope that this final comment allows you to re-evaluate the significance of our work.
>
> The authors
>
>
> ----
>
>
> [1] Mikulincer, Dan, and Daniel Reichman. "Size and depth of monotone neural networks: interpolation and approximation." Advances in Neural Information Processing Systems 35 (2022): 5522-5534.
>
> [2] Runje, Davor, and Sharath M. Shankaranarayana. "Constrained monotonic neural networks." International Conference on Machine Learning. PMLR, 2023.

---

> > ### Author Response · Authors · 2024-12-04
> > **Response to the edit of the official review**
> >
> > We would like to address the points raised in your recent edit to the official review, and clarify a few aspects of our rebuttal process:
> >  - **Notification of Edited Comments**: Please note that edits to previous comments are not automatically notified to recipients. The rebuttal process lasted three weeks, but we did not receive updates from you during the final two weeks until the most recent response, provided just hours before the deadline.
> >  - **Clarifications on Specific Points**:
> >    - Regarding our last comment, the pronoun issue (“his”) has been rectified. Our deepest apologies.
> >    - On the introduction of a constraint, we reiterate: **Bounded activations are simply a special case of alternating saturating activations**. This demonstrates a generalization of the result. Additionally, the **threshold activation has 0-gradient anywhere, rendering the previous result empirically impractical. Our approach, by contrast, employs activations with gradients, making it both theoretically and empirically valuable.**
> >    - On negative weights, we respectfully disagree with your assessment. Constrained MLPs typically utilize non-negatively constrained weights. Using **non-positively constrained weights is equally arbitrary but significantly more powerful, as evidenced by our theoretical results**.
> >    - On the optimization issues, Figures 5 and 6 **clearly demonstrate that the vanishing gradient dynamic arises due to initialization**. However, it is widely understood in the AI research community that sigmoid activations inherently lead to vanishing gradients, even with large step sizes. As such, it is unclear why we are required to present results that are already well-established and broadly recognized in the field. If you need a reference, feel free to consider [1, 2, 3, 4, 5, 6]
> >
> > We hope these clarifications help address your concerns.
> >
> > Sincerely,
> > The authors.
> >
> > ----
> >
> > [1] Dubey, Shiv Ram, Satish Kumar Singh, and Bidyut Baran Chaudhuri. "Activation functions in deep learning: A comprehensive survey and benchmark." Neurocomputing 503 (2022): 92-108.
> > [2] Ravikumar, Aswathy, and Harini Sriraman. "Mitigating Vanishing Gradient in SGD Optimization in Neural Networks." International Conference on Information, Communication and Computing Technology. Singapore: Springer Nature Singapore, 2023.
> > [3] Szandała, Tomasz. "Review and comparison of commonly used activation functions for deep neural networks." Bio-inspired neurocomputing (2021): 203-224.
> > [4] Nair, Vinod, and Geoffrey E. Hinton. "Rectified linear units improve restricted boltzmann machines." Proceedings of the 27th international conference on machine learning (ICML-10). 2010.
> > [5] Glorot, Xavier, and Yoshua Bengio. "Understanding the difficulty of training deep feedforward neural networks." Proceedings of the thirteenth international conference on artificial intelligence and statistics. JMLR Workshop and Conference Proceedings, 2010.
> > [6] Goodfellow, Ian, et al. "Maxout networks." International conference on machine learning. PMLR, 2013.

---

### Author Response · Authors · 2024-11-24

Dear Reviewers,

We sincerely thank you for your thoughtful and constructive feedback, as well as for the time and effort you have dedicated to reviewing our submission. Your comments have been invaluable in identifying areas for improvement and have guided us toward enhancing the clarity, rigor, and overall quality of our paper. Below, we provide a detailed account of how we have addressed your concerns and incorporated your suggestions.

**General Comments**
 - Benchmarks and Datasets: regarding the concern about the number and diversity of empirical results, we would like to emphasize that our experiments were conducted on benchmark datasets widely used in prior works [1,2]. These benchmarks were carefully selected to ensure comparability and relevance to the field. While we understand the desire for additional results, we believe that our chosen datasets offer a robust and representative evaluation of our approach.
 - Clarity and Presentation: we recognize the importance of clear communication and have undertaken significant revisions to improve the readability and presentation of the manuscript. Specific improvements are outlined below.

**Summary of Edits and Improvements**
 - **Revision of "Related Work"**: we rewrote this section to highlight the distinction between soft and hard monotonicity and to include citations for additional relevant work brought to our attention during the review process.
 - **Revision of theoretical work**: we revised our statement in Theorem 1 and addressed a missing term in the proof to ensure both correctness and clarity.
 - **Comprehensiveness of empirical comparison**: to improve the comprehensiveness of our empirical comparisons, we incorporated a recently published work that we had initially overlooked, ensuring a fairer and more complete evaluation.
 - **Study on optimization issues**: we added a focused study on a toy scenario in the appendix, illustrating the optimization dynamics of non-negatively constrained monotonic MLPs with saturating activations and those with ReLU activations. We did not include a study on ReLU6 since the same MLPs with such activation leads to all gradients to be zero ($\frac{df_\theta(x)}{d\theta}=0$). We hope this addition provides valuable insights into the practical challenges and trade-offs.
 - **Overall writing revision**: we addressed the typos and overly long phrases highlighted by reviewers. Additionally, we conducted a thorough revision of the manuscript to rectify other stylistic and grammatical issues, ensuring that the paper is more concise and readable.
 - **Ethical statement**: we also included an ethical statement in the manuscript, as suggested by one of the reviewers, to address concerns regarding the use of the COMPAS dataset. This addition reflects our commitment to acknowledging and discussing the ethical implications associated with our work.

 In conclusion, we deeply appreciate the constructive dialogue that has been taking place so far during this rebuttal process. We sincerely hope that our detailed responses and the substantial revisions we have made demonstrate our commitment to addressing your feedback thoughtfully and comprehensively. We hope these efforts meet your expectations and encourage you to reassess your evaluation of our submission positively.

Thank you once again for your time, effort, and valuable insights.
Best regards,
Authors



----

References:

[1] Kim, Hyunho, and Jong-Seok Lee. "Scalable Monotonic Neural Networks." The Twelfth International Conference on Learning Representations.

[2] Runje, Davor, and Sharath M. Shankaranarayana. "Constrained monotonic neural networks." International Conference on Machine Learning. PMLR, 2023.

---

### Note · Authors · 2025-01-23

I have read and agree with the venue's withdrawal policy on behalf of myself and my co-authors.